# Acoustic spin rotation in heavy-metal-ferromagnet bilayers

Yang Cao[1], Hao Ding[1], Yalu Zuo[1], Xiling Li[1], Yibing Zhao[1], Tong Li[1], Na Lei [2], Jiangwei Cao[1], Mingsu Si[1], Li Xi [1], Chenglong Jia [1], Desheng Xue[1] ✉ & Dezheng Yang [1] ✉

Through pumping a spin current from ferromagnet into heavy metal (HM) via magnetization precession, parts of the injected spins are in-plane rotated by the lattice vibration, namely acoustic spin rotation (ASR), which manifests itself as an inverse spin Hall voltage in HM with an additional 90° difference in angular dependency. When reversing the stacking order of bilayer with a counter-propagating spin current or using HMs with an opposite spin Hall angle, such ASR voltage shows the same sign, strongly suggesting that ASR changes the rotation direction due to interface spin-orbit interaction. With the drift-diffusion model of spin transport, we quantify the efficiency of ASR up to 30%. The finding of ASR endows the acoustic device with an ability to manipulate spin, and further reveals a new spin-orbit coupling between spin current and lattice vibration.

Recent advances in spin-acoustics have renewed interest in pure spin current phenomena, involving the interaction between the lattice vibration and the spin-charge dynamics of electrons[1]. Owing to the magnetoelastic effect, lattice vibration-induced time-varying strain enables ferromagnetic resonance (FMR)[2–5], thus producing a pure spin current in ferromagnet (FM)[6]. When further considering the lattice rotation, the pure spin current can also be produced in nonmagnetic light metal (LM), by converting the mechanical angular momentum of the elastic wave into the spin angular momentum of electron[7–10]. The lattice chiral rotation induced by SAW is also responsible for non-reciprocal phenomena in the acoustic device[5,11–14], and leads to possibilities to sense the magnetic chiral structure[15,16]. Moreover, in analogy to the spin Hall effect (SHE)[17–20], where the role of charge current is replaced by the elastic wave, the pure spin current even emerges in heavy metal (HM) from the dynamics of lattice due to the spin-orbit interaction (SOI)[21]. These pioneering works have well established the acoustic generation of pure spin current in both FM and nonmagnet (NM), however, there is a hidden assumption that the spin is conserved during the electron-lattice scattering in acoustic spin devices.

In fact, with SOI breaking the spin-rotation symmetry[22], spin is never conserved. Under the scenario of electron-lattice scattering, the spin-orbit field of lattice not only spin-selectively compels the electrons to move into opposite transverse directions, thus resulting in a pure spin current, i.e., SHE[17–20], but also can rotate the spin direction via the spin precession[23–27]. Recent studies have demonstrated that an additional spin current with spin rotation due to SOI is developed in the low-symmetry point group crystals, such as LaFeO$_3$[27], WTe$_2$[28], MoTe$_2$[29], CuPt[30], Mn$_2$Au[31], Mn$_3$GaN[32], Mn$_3$SnN[33], Mn$_3$Sn[34], and so on. Because of its relationship with charge-spin conversion and lattice symmetry, spin rotation provides fundamental insight into the mechanism of spin-lattice scattering. Besides the bulk effect in crystals, spin rotation can also occur at the FM/NM interface, which results from structure inversion asymmetry of the confinement potential (Rashba SOI)[35–40]. Since the spin direction is the key parameter of torques that the spin current exerts on FM, spin rotation has been used to control the magnetization dynamics, leading to a much more efficient and deterministic magnetization switching of nanomagnets[28,32,41,42], in particular to the field-free switching of a perpendicular magnetization[30,31,43,44]. So far, acoustic spin rotation (ASR) has not been reported in the literature. Distinct from the existing electric spin devices that use charge motion to control spin state, such acoustic spin rotation uses lattice motion as a new degree of freedom to realize

[1]Key Laboratory for Magnetism and Magnetic Materials of Ministry of Education, Lanzhou University, Lanzhou 730000, China. [2]Fert Beijing Institute, MIIT Key Laboratory of Spintronics, School of Integrated Circuit Science and Engineering, Beihang University, Beijing 100191, China. ✉e-mail: xueds@lzu.edu.cn; yangdzh@lzu.edu.cn

spin rotation, which causes noncontact and more efficiently controls of the spin bits in spintronic devices and in quantum computing[45].

To experimentally observe ASR, we chose surface acoustic wave (SAW)-induced magnetization precession to generate a pure spin current and detected its amplitude and spin direction in HM with the aid of inverse SHE (ISHE)[19,46]. The advantage of the acoustic spin pumping (ASP) method is that the spin direction of the injected spin current can be freely controlled by rotating the magnetic field. This allows us to check ASR for injecting any spin direction, and to exclude the thermal and microwave effects due to their different angular dependence. As shown in Fig. 1a, if the injected spin $\sigma$ is unchanged, it will induce an ISHE voltage, namely ASP voltage, while if $\sigma$ has a probability to be rotated 90° to $\sigma'$ via the spin-lattice scattering, it will produce an additional ISHE voltage with an exact 90° difference in angular dependency from ASP voltage, namely ASR voltage, as shown in Fig. 1b. The 90° difference in angular dependency between the ASP and ASR voltages can provide strong evidence of ASR in FM/HM bilayer.

In this work, we observe ASR from the dynamic of lattice via SOI in HM/FM bilayers. We show that the ASP and ASR voltages coexist in Pt/Ni bilayer, and have the same magnetic field and SAW dependence. The observed exact 90° difference between the ASP and ASR angular dependency strongly suggests spin is rotated with a fixed probability by lattice vibration, for injecting any spin direction. Measurements of the bilayers with reversed stacking order, i.e., Ni/Pt and Ni/Ta bilayers, further show the ASP and ASR voltages are respectively proportional to spin Hall angle ($\theta_{SH}$) and $\theta_{SH}^2$. By perfectly fitting the ASP and ASR voltages with the drift-diffusion model of spin transport, we find that the conversion efficiency of spin rotation in acoustic devices scales with the interface SOI, independent of the magnetic field and SAW. Finally, we discuss the mechanism of ASR by using a unified picture of spin rotation under the out-of-plane spin-orbit field due to lattice vibration.

## Results

### The FMR driven by SAW

Two interdigital transducers (IDTs) on 128° Y-cut LiNbO$_3$ substrates are used to launch and detect the SAW, as shown in Fig. 2a. IDT1 is designed as a stepped-finger IDT[47], which can enhance the 5$^{th}$ harmonic while suppressing all other harmonics. The finger width and spacing are both 4.25 $\mu$m, and the distance of the two IDTs is 600 $\mu$m. The propagation direction of SAW is along the X-crystalline axis of the LiNbO$_3$ substrate, which is defined as $x$-axis. Along this propagation direction, the SAW is Rayleigh-mode, which means that the vibration is only along the $x$ and $z$ directions. Therefore, there are three

nonvanishing strain components at the surface of the substrate, namely, $\varepsilon_{xx}$, $\varepsilon_{zz}$, and $\varepsilon_{xz}$. In acoustic device, $\varepsilon_{xx}$ dominates the magnetization dynamics[3] while $\varepsilon_{xz}$ leads to nonreciprocity[11]. The NM/FM bilayer (or the FM/NM bilayer with reversed stacking order) was deposited between the two IDTs by magnetron sputtering, and patterned to a Hall bar by optical lithography. The device structure is LiNbO$_3$/Ti(5)/NM(2)/Ni(30)/Ti(3) with NM = Cu, Pt, and Ta (thickness in nanometers). All samples have in-plane magnetization. By launching SAW on one IDT, we measured both the SAW transmission on the other IDT and the longitudinal ISHE voltage $V_{xx}$ of the Hall bar. The magnetic field $H$ is rotated within the $xy$ plane to obtain the angular dependence of both the SAW transmission and $V_{xx}$.

Figure 2b shows the typical transmission spectra of the SAW device measured by a vector network analyzer. The time-domain gating technique is used to cancel the contributions of electromagnetic crosstalk[48,49]. As we expected, besides the fundamental frequency at 0.22 GHz, there is also a strong 5$^{th}$ harmonic at 1.10 GHz. Owing to the large magnetoelastic interaction of Ni, an acoustic FMR (AFMR)[2-4] of Ni film is excited by the 5$^{th}$ harmonic SAW, as shown in Fig. 2c. The resonance field of Pt/Ni bilayer is around 4 mT, and the peaks at $\mu_0H = \pm 2$ mT are due to the magnetization reversal of Ni. Figure 2d shows that the S$_{21}$ curves for magnetic fields $H$ from 4 to 18 mT all exhibit the sin$^2$2$\varphi$ angular dependence. This is the fingerprint of acoustic ferromagnetic resonance that distinguishes it from microwave-field induced ferromagnetic resonance[2]. As $H$ is away from the acoustic ferromagnetic resonance, the power absorption S$_{21}$ gradually decreases.

### Symmetric analysis to extract the ASR voltage

Angular-dependent measurements of $V_{xx}$ for Pt/Ni bilayer offer insight into the different symmetric relationships of the ASP and ASR. Under mirror reflection in the $xz$ plane $\sigma_{xz}$, only the in-plane magnetization **m** is changed from $\varphi$ to $180° - \varphi$, since **m** is a pseudovector, thus ASP voltage should exhibit a symmetric angular dependence with respect to $\varphi = 90°$, i.e., $V_{ASP}(180° - \varphi) = V_{ASP}(\varphi)$, whereas ASR should exhibit an antisymmetric angular dependence, i.e., $V_{ASR}(180° - \varphi) = -V_{ASR}(\varphi)$, because the direction of in-plane spin rotation is inverted under $\sigma_{xz}$. As shown in Fig. 3a, the angular dependence of $V_{xx}$ for Pt/Ni bilayer exhibits significant asymmetric behavior with respect to $\varphi = 90°$. To demonstrate the asymmetry $V_{xx}$ originates from ASR, we decompose $V_{xx}$ into two components that are symmetric $V_{xx}^S$ [Fig. 3b] and antisymmetric $V_{xx}^A$ [Fig. 3c] with respect to $\varphi = 90°$. The angular dependence of $V_{xx}^S$ is the same as the previous ASP results[6], which reads

$$V_{xx}^S(\varphi) = V_{ASP} \sin\varphi \sin^2 2\varphi, \tag{1}$$

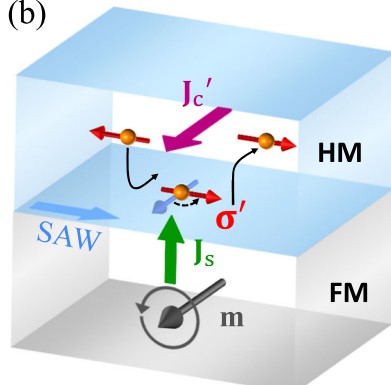

**Fig. 1 | Illustration of the acoustic spin pumping (ASP) and acoustic spin rotation (ASR) in ferromagnet (FM)/heavy metal (HM) bilayer.** A schematic diagram of **a** ASP and **b** ASR in FM/HM bilayer. The initial spin direction $\sigma$ of the injected spin current **J**$_s$ is along the magnetization **m** due to spin pumping. **a J**$_s$ is

directly converted to charge current **J**$_c$ or voltage $V_{ASP}$ in HM via inverse spin Hall effect (ISHE). **b** Due to surface acoustic wave (SAW)-induced lattice vibration, the injected spin $\sigma$ is rotated 90° to $\sigma'$, which is converted to an additional charge current **J**$_c'$ or voltage $V_{ASR}$ in HM via ISHE.

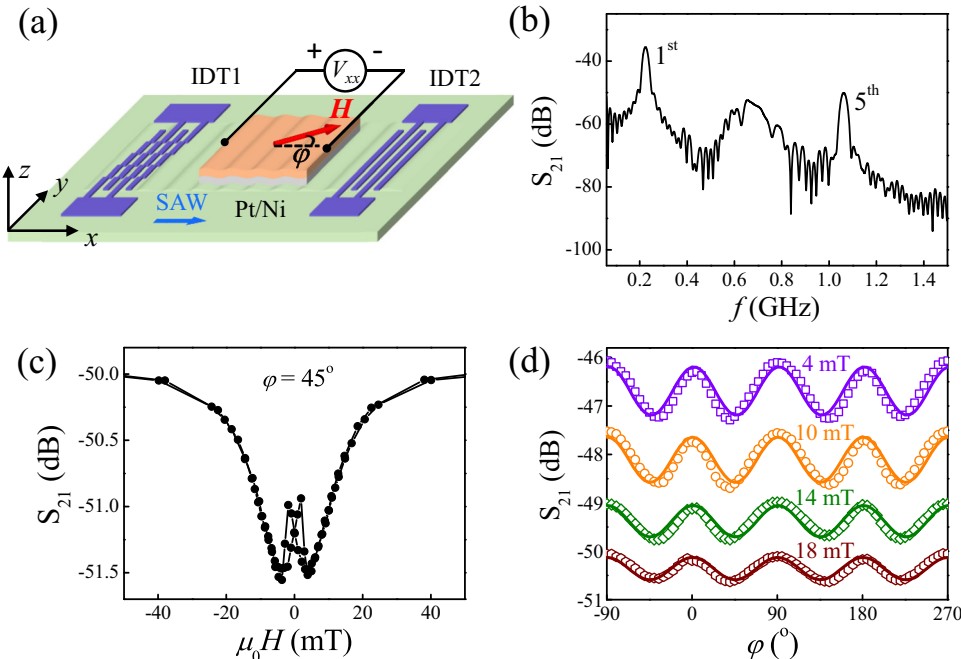

**Fig. 2 | Experimental setup and acoustic FMR (AFMR) of Pt/Ni bilayer.**
**a** Schematic of the SAW device and measurement configuration. **b** The $S_{21}$ transmission parameter of the SAW device as a function of $f$. **c** The field dependence of

$S_{21}$ under the 5th harmonic SAW that excites AFMR of Pt/Ni bilayer. **d** The angular dependence of $S_{21}$ for Pt/Ni under the 5th harmonic SAW for different $H$.

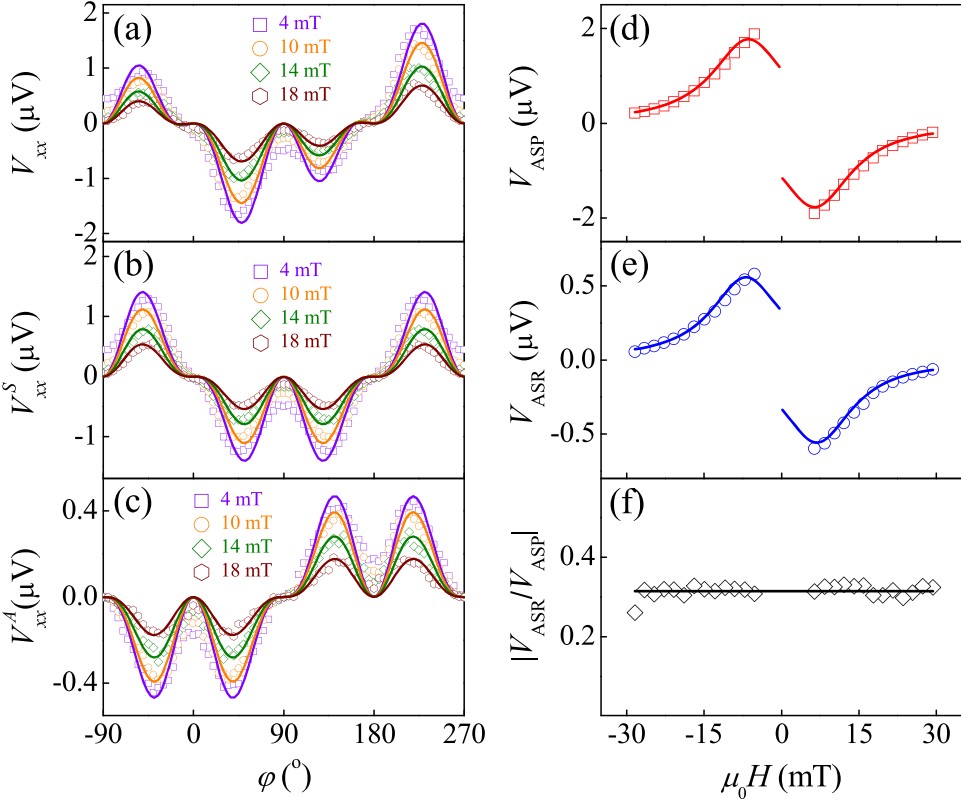

**Fig. 3 | Separation of ASR and ASP from ISHE voltage based on symmetry. a** The angular dependence of $V_{xx}$ for Pt/Ni under the 5th harmonic SAW excitation with $P =$ 10 mW for different $H$. We extract $V_{xx}$ into **b** the symmetric ASP voltage and **c** the

antisymmetric ASR voltage with respect to $\varphi = 90°$. The $H$ dependence of **d** ASP voltage ($V_{ASP}$), **e** ASR voltage ($V_{ASR}$), and **f** $|V_{ASR}/V_{ASP}|$ for Pt/Ni bilayer. The solid lines represent the Lorentz fit with respect to $H$.

where $V_{ASP}$ represents the amplitude of ASP voltage. Remarkably, one can find there is exactly a 90° difference in angular dependency between $V_{xx}^A$ and $V_{xx}^S$. By shifting a 90° phase of $V_{xx}^S$, we directly get the angular dependence of $V_{xx}^A$

$$V_{xx}^A(\varphi) = V_{ASR}\cos\varphi\sin^2 2\varphi, \qquad (2)$$

where $V_{ASR}$ represents the amplitude of ASR voltage. We attribute the angular dependence of $V_{xx}^A$ to the ASR for the following reasons. Firstly, the 90° difference in angular dependency between $V_{xx}^A$ and $V_{xx}^S$ agrees well with the picture of the 90° spin rotation. Secondly, since the injected spin current amplitude is caused by AFMR [Fig. 2d], this enters $V_{xx}^A \propto \sin^2 2\varphi$. The rest contribution of the $\cos\varphi$ dependence is due to the projection of $\boldsymbol{\sigma}'$ induced ISHE voltage along the $x$-direction. Thirdly, we further measure the angular dependence of the Hall voltage of ASR, which still obeys with our picture. When considering the $\sin\varphi$ projection of $\boldsymbol{\sigma}'$ induced ISHE voltage along the $y$-direction (Hall direction), the Hall voltage of ASR should be a $\sin\varphi\sin^2 2\varphi$ angular dependence, which is confirmed by experiments in Section E of the Supplemental Material. Finally, this anomalous signal is not derived from microwave induced rectification voltage[5]. By using $SiO_2$ to block the spin current from Ni to Pt, we found both $V_{ASP}$ and $V_{ASR}$ vanish, as shown in Fig. S6 in Section G of the Supplemental Material. This suggests $V_{ASP}$ and $V_{ASR}$ both originate from the pumping spin current. Besides $V_{ASP}$ and $V_{ASR}$, we didn't observe any rectification signal when $SiO_2$ is inserted.

After fitting the angular dependence of $V_{xx}$, we obtain the $H$ dependence of $V_{ASP}$ and $V_{ASR}$ in Fig. 3d and 3e, respectively. The $V_{ASP}$ and $V_{ASR}$ show the same field dependence. The maximum and minimum values both occur at the resonance field. As $H$ is away from AFMR, the amplitudes of $V_{ASP}$ and $V_{ASR}$ gradually approach 0. Since the magnetization is not uniform and hysteretic in the low field region, we only measure fields $|\mu_0 H| \geq 4$ mT. Because of the resonance, the $H$ dependence of $V_{ASP}$ and $V_{ASR}$ are fitted very well by the symmetric Lorentz-lineshape $L(H)$, which reads

$$V_{ASP} = c_{ASP}\text{Sign}(H)L(H), \qquad (3)$$

$$V_{ASR} = c_{ASR}\text{Sign}(H)L(H), \qquad (4)$$

where $c_{ASP/ASR}$ depends on the material properties and the SAW frequency (See Eqs. (S33) and (S38) in the Supplemental Material). We did not observe the microwave induced antisymmetric Lorentz lineshape[50] in $V_{ASP}$, $V_{ASR}$ as well as raw $V_{xx}$ in Fig. S9 in Section I of the Supplemental Material. This also suggests the microwave-induced rectification voltage[5] is negligible in our system. We notice that the field dependence of $V_{ASP}$ and $V_{ASR}$ are odd. This again supports the $V_{ASP}$ and $V_{ASR}$ both originate from the injected spin current. Because the spin direction of the injected spin current is changed under magnetization reversal, this induces the sign change of ISHE voltage. The same field dependence of the $V_{ASP}$ and $V_{ASR}$ not only demonstrates that they are derived from SAW-induced spin pumping, but also indicates that the conversion efficiency $\eta = |V_{ASR}/V_{ASP}|$ is independent of $H$. We show that $\eta$ for Pt/Ni bilayer is up to 30% in Fig. 3f, which is even larger than the values in electric spin devices[23,27].

Further support for $V_{ASP}$ and $V_{ASR}$ arising from the injection of spin current comes from the SAW frequency and power dependence shown in Fig. 4. A complete Lorentz-lineshape in a narrow frequency range has been observed for both ASP and ASR, as shown in Fig. 4a and b respectively. When the frequency deviates from the 5th harmonic frequency of our SAW device, the lattice vibration is strongly reduced, leading to a decrease of the injected spin current due to the reduction of magnetization precession. The sharply resonant also rules out the thermal and microwave effects. We further find that the $V_{ASP}$ and $V_{ASR}$ are both proportional to the power of SAW ($P$), as shown in Fig. 4c. This is consistent with the previous ASP results that $J_s \propto \varepsilon_{xx}^2 \propto P^6$. The same SAW dependence for ASP and ASR also yields $\eta = 0.3$ for Pt/Ni bilayer, independent of SAW in Fig. 4d.

## Evidence to confirm ASR originates from the interface SOI

To directly demonstrate ASR is due to SOI, we first fabricated a Ni/Pt bilayer control sample with reversed stacking order, and also measured both the $V_{ASP}$ and $V_{ASR}$. The reversed stacking order makes the spin current $\mathbf{J}_s$ counter-propagate in Fig. 5a, thus producing an opposite ISHE voltage in Pt. Compared to the Pt/Ni bilayer in Fig. 4a, we observe the $V_{ASP}$ of Ni/Pt bilayer indeed reverses sign in Fig. 5b, indicating that spin current has been reversely injected into Pt. However, the sign of $V_{ASR}$ remains unchanged, suggesting that the reversed stacking order further changes the direction of spin rotation. Next, we further show that ASR is related to SOI, by replacing Pt with Ta. Due to the opposite $\theta_{SH}$ shown in Fig. 5a and 5c, $V_{ASP}$ changes signs, whereas

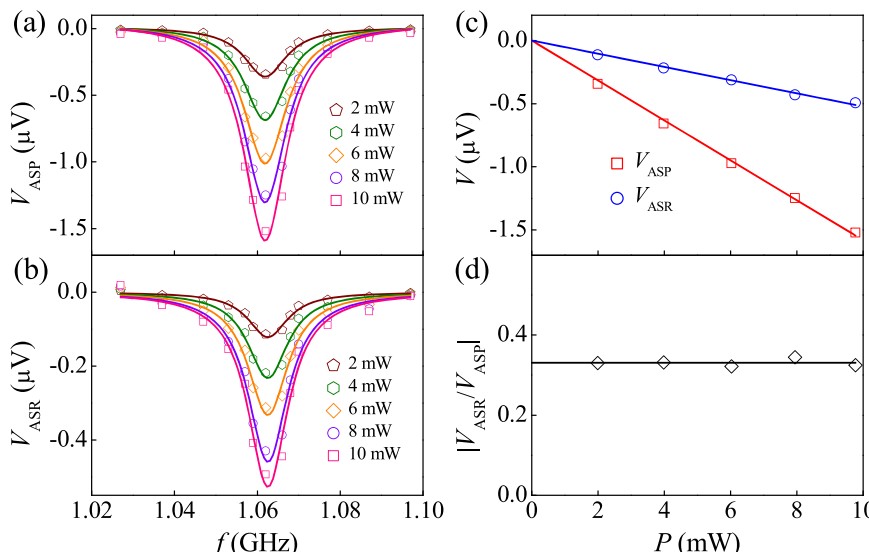

**Fig. 4 | Frequency and power dependence of the ASP voltage and ASR voltage.** The $f$ dependence of **a** $V_{ASP}$ and **b** $V_{ASR}$ for Pt/Ni bilayer for different SAW powers $P$. The $P$ dependence of **c** $V_{ASP}$, $V_{ASR}$ and **d** $|V_{ASR}/V_{ASP}|$ for Pt/Ni bilayer at the 5th harmonic frequency.

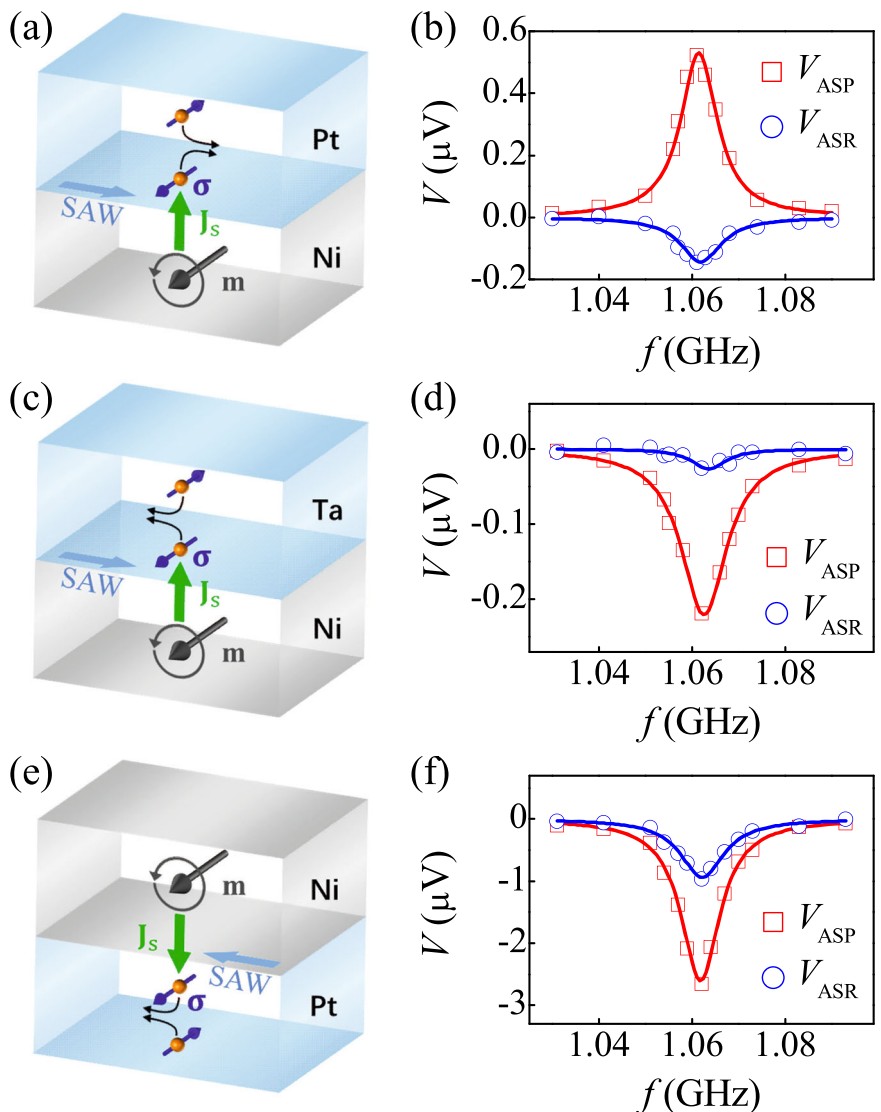

**Fig. 5 | ASP voltage and ASR voltage of the control samples.** Schematic diagram of ASP and ASR, and the $f$ dependence of $V_{ASP}$ and $V_{ASR}$ with $P = 10$ mW in **a**, **b** Ni/Pt, **c**, **d** Ni/Ta, and **e**, **f** Pt/Ni, respectively.

$V_{ASR}$ still keeps the same sign, as shown in Fig. 5d. The same sign of $V_{ASR}$ implies the direction of spin rotation changes again, indicating that ASR arises from the SOI. Besides, we noted that the $V_{ASR}$ for Ni/Ta bilayers is smaller than that for Ni/Pt bilayers. This may be due to the shorter spin relaxation time or spin diffusion length of Ta[51]. As a result, the spin has been scattered before rotation. Finally, we point out that the antisymmetric acoustic voltage observed here is not caused by SAW attenuation. The SAW attenuation along the propagation direction will also induce an antisymmetric acoustic voltage in strong Rashba system Ni/Cu/Bi$_2$O$_3$ and Ni/Ag/Bi$_2$O$_3$ hybrid devices[52], which changes signs with the SAW propagation direction. When we reverse the SAW propagation as shown in Fig. 5e, we find that the sign of antisymmetric component for Pt/Ni bilayer in Fig. 5f is still the same. This indicates that the antisymmetric acoustic voltage in our case is mainly caused by ASR. In addition, compared with the results in Fig. 4 (+$x$ propagation), both $V_{ASP}$ and $V_{ASR}$ in Fig. 5f (-$x$ propagation) are enhanced. We attribute it to both the nonreciprocity of system[5,11–14] and the difference between two IDTs.

## Discussion

We ascribe the experimental results to the interface SOI-induced spin-orbit field $\mathbf{B}_{SO}$. Since the additional ISHE voltage always has a 90°

difference in angular dependency with the ASP voltage for injecting any in-plane spin, there should be a $z$-direction spin-orbit field $\mathbf{B}_{SO}$ to rotate the in-plane spin $\boldsymbol{\sigma}$ via the spin precession, which reads

$$\mathbf{J}_s^{ASR} = \alpha J_s \mathbf{B}_{SO} \times \boldsymbol{\sigma}, \tag{5}$$

where $J_s$ is the injected spin current with spin $\boldsymbol{\sigma}$, $\mathbf{J}_s^{ASR}$ is the ASR-induced spin current with spin direction along $\mathbf{B}_{SO} \times \boldsymbol{\sigma}$, and $\alpha$ is a coefficient. Similar to the spin rotation in the electric spin device, the acoustic spin rotation origins from the spin precession around $\mathbf{B}_{SO}$. For example, when applying an electric field to drive electron motion in the FM/HM system, the charge current can be converted into two polarized spin currents due to the spin-orbit coupling. One is the conventional $y$-polarized spin current. The other is anomalous $z$-polarized spin current, which has been widely reported[28–36,43]. Amin et al. describe such anomalous $z$-polarized spin current to the spin-orbit precession effect around the $y$-direction interface Rashba field[36]. Due to the spin precession effect, the injected $\boldsymbol{\sigma}$ polarization spin current can be rotated around $y$-direction interface Rashba field to $\boldsymbol{\sigma}'$ polarization spin current, i.e., $\boldsymbol{\sigma}' = \mathbf{y} \times \boldsymbol{\sigma}$, where $\boldsymbol{\sigma}'$ is the spin direction of spin current under spin precession, $\mathbf{y}$ is the direction of the interface Rashba field, and $\boldsymbol{\sigma}$ is the spin direction of the injected spin current. For

in-plane ferromagnetic magnetization **m**, and $\sigma \| \mathbf{m}$, the spin-orbit precession will cause a $z$-polarized spin current. Because in our case the effective spin-orbit field is along $z$-direction, i.e., $\sigma' = \mathbf{z} \times \sigma$, as a result, when injecting in-plane $\sigma$ spin current, we will find the $\sigma'$ due to spin rotation is always exactly 90°.

Below we explain that $\mathbf{B}_{SO}$ originates from the SAW attenuation in the $z$-direction. Since in our case, the acoustic wave is a Rayleigh surface wave, the lattice displacements decay exponentially along the $z$-direction. When SAW acts on the FM due to magnetoelastic coupling, the dynamic strain due to lattice displacement causes the magnetization precession. As shown in Fig. 6a, the $z$-direction SAW attenuation will lead to the decay of the magnetization precession. As a result, the time-averaged equivalent magnetization **M** also decays along the $z$-direction, which causes a gradient of magnetization along the $z$-direction. The equivalent magnetization during magnetization precession can be described by $\mathbf{M} = \mathbf{M}_s e^{-z/\lambda}$, where $M_s = 4.85 \times 10^5$ A/m is the saturated magnetization of Ni, $z = 30$ nm is the thickness of Ni, and the attenuation depth of SAW $\lambda$ is close to the wavelength of SAW[21], i.e., $\lambda \approx 17 \mu m$. According to Zhang's theoretical work, the nonuniform distribution of magnetization can produce an additional torque (also called Zhang-Li torque)[53], which has been verified by experiments[54–56]. Owing to the $z$-direction magnetization decay induced by SAW, the Zhang-Li torque here can be expressed as $\tau = -(c/M_s)\mathbf{M} \times (\partial M/\partial z)\mathbf{z}$. In our case, the magnetization gradient along the $z$-direction $\partial M/\partial z = (M_s/\lambda)e^{-z/\lambda} = 2.9 \times 10^{10}$ A/m$^2$. It should be noted the magnetization gradient induced by SAW is four orders of magnitude larger than that induced by electrical field at PMN-PT/Pt interface[54]. The corresponding $z$-direction spin-orbit field can be expressed as $\mathbf{B}_{SO} = -(c/M_s)(\partial M/\partial z)\mathbf{z}$. The coefficient $c$ depends on effective interfacial spin Hall angle $\theta_{SH}^{eff}$, and is proportional to the intrinsic $\theta_{SH}$. If we use $c = 4.8 \times 10^{-5}$ Tm for Pt/Co/Ni/Co/Pt system[54], we can obtain $B_{SO} = -2.9$ T, which is large enough to induce the acoustic spin rotation.

Because $\mathbf{B}_{SO}$ and the following ISHE voltage in HM both depend on $\theta_{SH}$, $V_{ASR}$ is proportional to $\theta_{SH}^2$, thus keeping the same sign for both Pt and Ta. For bilayer with reversed stacking order, $\mathbf{B}_{SO}$ is reversed from $z$ to $-z$, thus reversing the direction of spin rotation. When we further consider spin drift and diffusion by using the Landau-Lifshitz-Gilbert (LLG) equation in the Supplemental Material, we obtain all the field and SAW dependencies of $V_{ASP}$ and $V_{ASR}$, which consist with our measurement results.

To prove $\mathbf{B}_{SO}$ originates from the SAW attenuation, we studied the dependence of acoustic spin rotation efficiency for Pt(2)/Ni(30) bilayer on the SAW frequency. Due to the SAW attenuation in the $z$-direction, the equivalent magnetization during magnetization precession also decays in the form of $\mathbf{M} = \mathbf{M}_s e^{-z/\lambda}$. Since the attenuation depth $\lambda$ is inversely proportional to SAW frequency $f$[21], the gradient of

magnetization $\partial M/\partial z = (M_s/\lambda)e^{-z/\lambda} \approx M_s/\lambda \propto f$. Here we consider $e^{-z/\lambda} \approx 1$, because the thickness of the ferromagnetic layer $z$ is in nanometer scale, while $\lambda$ in our case is about several $\mu m$. As shown in Fig. 6b, $\eta$ is almost proportional to the SAW frequency, further proving the correction of the acoustic spin rotation model.

In conclusion, besides the symmetric ASP voltage, we observe an antisymmetric ASR voltage in HM/FM bilayers under SAW-induced magnetization precession. We show that the ASP and ASR voltages exhibit the same Lorentz field dependence and the same linear acoustic power dependence, but always have a 90° difference in angular dependency for pumping spin current with any spin direction. These results can be described by our drift-diffusion model of spin transport. Another significant difference is that ASP voltage is proportional to $\theta_{SH}$, whereas ASR voltage is proportional to $\theta_{SH}^2$, which provides important evidence that ASR originates from the interface SOI-mediated coupling of electron spins and the lattice. Unlike the previous acoustic spin effects that focused on the generation of spin current, the finding of ASR provides insight into how to manipulate a spin current in the acoustic device, which will invigorate the spin-acoustics.

## Methods

### Device fabrication

Magnetron sputtering is used to deposit the films on piezoelectric LiNbO$_3$ substrates. The base pressure of magnetron sputtering is below $5.0 \times 10^{-5}$ Pa. The film structure is LiNbO$_3$/Ti(5)/HM(2)/Ni($d$)/Ti(3) with HM = Pt and $d = 5, 10, 20, 30, 50$ (thickness in a unit of nanometers). The inverted structure is LiNbO$_3$/Ti(5)/Ni(30)/HM(2)/Ti(3) with HM = Pt and Ta. The Ti(5) layers serve as a buffer layer while the Ti(3) layers serve as a capping layer to prevent oxidation of the films. The films are patterned to Hall bar by optical lithography and Ar ion etching. The length and width of the Hall bar are set to 510 $\mu$m and 290 $\mu$m, respectively.

Subsequently, we use optical lithography and a liftoff process to form IDTs and electrodes made of Ti(5)/Au(70). IDT1 is designed as stepped-finger to enhance the 5$^{th}$ harmonic. The distance of the two IDTs is 600 $\mu$m and each IDT has 20 pairs of fingers. We set the finger widths and spacings of IDTs to 4.25, 2.87, 2.13, 1.88, and 1.43 $\mu$m to obtain the different SAW frequencies. The SAW frequencies are 1.06, 1.70, 2.29, 2.60, and 2.05 GHz, where 2.05 GHz is the third harmonic and the other frequencies are the fifth harmonic.

### Measurements of SAW transmission and ISHE voltage

The SAW transmission was measured with a vector network analyzer. The longitudinal (along $x$) and transverse (along $y$) ISHE voltages, defined as $V_{xx}$ and $V_{xy}$, respectively, are measured during the SAW excitation.

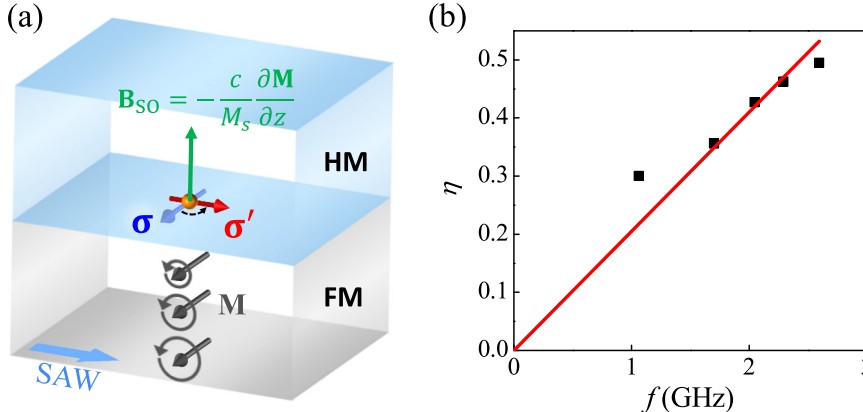

**Fig. 6 | Microscopic picture of ASR. a** Schematic diagram of the physical origin of ASR. **b** ASR efficiency $\eta = |V_{ASR}/V_{ASP}|$ of SAW devices with different frequencies (i.e., different finger widths).

## Data availability

The data that support the findings of this study are available from the corresponding authors upon reasonable request.

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

## Acknowledgements

This work is supported by the NSFC of China (Grant Nos. 11774139, 11874189, 91963201, 12074025, and 52061135105), PCSIRT (Grant No. IRT-16R35), Fundamental Research Funds for the Central Universities lzujbky-2021-ct01, and the 111 Project under Grant No. B20063.

## Author contributions

Y.C. and D.Y. conceived the idea and designed the experiment. Y.C., H.D., X.L., Y.Z. and T.L. prepared the samples. Y. C. performed the measurements. L.Z., J.C., N.L., M.S., L.X., C.J. and D.X. gave suggestions on the experiments and theory. All authors contributed to discussions. Y.C. and D.Y. analysed the data and wrote the manuscript.

## Competing interests

The authors declare no competing interests.
