## [Peer Review File · Nature Communications]

Reviewers' Comments:

Reviewer #1:

Remarks to the Author:

This paper experimentally demonstrated that an injection of surface acoustic waves into the interface between ferromagnet (FM) and heavy metal (HM) with a strong SOI generates an effective magnetic field B_{so} perpendicular to the interface, whose polarity depends on the sign of the spin Hall angle of the HM. This effective magnetic field causes the precession of electron spin at the interface, resulting in a change in the polarization direction of the spin current pumped from the FM. The authors measured the field-angle-dependent inverse spin Hall voltage generated in the x-y plane bilayer by injecting surface acoustic waves (SAWs) in the x-axis direction. The results showed that both symmetric and antisymmetric components with respect to the mirror inversion operation in the y-z plane were generated simultaneously. The authors claimed that the origin of the antisymmetric component was the acoustic spin rotation (i.e., ASR). To show that the ASR is due to the interface SOI, it was demonstrated that the antisymmetric component does not change its sign when the stacking order of the bilayer is reversed or the sign of the spin Hall angle is reversed. Namely, the spin rotation direction is reversed after reversing the stacking order or the sign of the spin Hall angle. Since the sign of the interface SOI is generally reversed by the direction of broken spatial inversion symmetry or the sign of intrinsic SOI, the authors' claim seems reasonable.

The argument based on the analysis of the experimental methods is systematically presented and the quality of the paper seems to be high. However, there are several concerns, and it cannot be recommended for publication in the present form.

(1) The authors claim that the magnetization gradient in the z-direction induced by SAWs produces an effective magnetic field B_{so} . If this is true, to clarify the origin of B_{so} , the authors should examine the dependence of the ASR signal on both the thickness of the FM layer and the frequency of the SAWs. The SAW attenuation depth along the z-direction decreases with increasing the frequency. Namely, the strength of B_{so} should be increase with the frequency.

(2) The crystal cut surface of the lithium niobate (LN) substrate used in this study must be specific. Furthermore, the authors should describe which mode of SAWs is excited and which strain tensor components contribute to the magnetic dynamics. For instance, in the case of a 128-degree Y-cut LN substrate, it is known that the phase of the field angle variation of the inverse spin Hall voltage possibly depends on the crystalline orientation of the SAW propagation.

(3) Does the dimension of the formula, which is given in the present paper for the effective interfacial torque due to the effective magnetic field, is the torque per unit area? I believe that the time derivative of magnetization should be included in the formula for torque. If so, the effective magnetic field should depend on frequency. Demonstrating the frequency dependence of the ASR effect is, therefore, very important for understanding the physics.

(4) It should be explicitly commented how the discovery of the ASR effect contributes to device applications. This is crucial for assessing the level of acceptance for publication in Nature Communications.

(5) What intensity of microwaves was used in the experiment in Fig. 5? Comparing the results in Fig. 4 and Fig. 5(f), both the ASP and ASR signals are enhanced when SAWs propagate in the negative-x direction. Is this effect quantitatively explained by the non-reciprocal propagation characteristics of surface acoustic waves in Ni thin films?

Reviewer #2:

Remarks to the Author:

The authors experimentally study the dc voltage arising in a Ni/Pt bilayer when exposed to a surface acoustic wave (SAW). Previous studies have identified the acoustic spin pumping (acoustically driven magnetization precession + spin pumping + inverse spin Hall effect) as the origin of the dc voltage. Here, the authors report that the symmetry of the measured voltage with

respect to the orientation of the external magnetic field direction does not match the expectations from spin pumping/iSHE alone. In the spin pumping scenario, the expected symmetry would be $\sin(\phi)$, where ϕ is the angle between the voltage contacts and the magnetization. The authors find an additional contribution to the dc voltage under the condition of magnetic resonance with a $\cos(\phi)$ dependence. The authors attribute this voltage component to a 90° spin rotation at the Ni/Pt interface and subsequent detection by the iSHE in Pt. Control experiments with Ni/Ta and Pt/Ni samples are performed and show that the symmetry of the additional voltage component is independent of stacking order, nonmagnetic metal and SAW propagation direction.

The topic is certainly relevant and timely, and the experimental data clearly indicates a contribution to the dc voltage that cannot be explained in the context of the established model of acoustic spin pumping. However, I found that the interpretation of the authors in terms of a so-called acoustic spin rotation is not supported by the data as explained in the following.

Because the conclusions are not supported by the data, the manuscript cannot be published. I recommend to either submit the data without accompanying model/interpretation to an appropriate outlet (such as scientific reports) for further exploration by the community or to perform a much more substantial study to clarify the actual origin of the unexpected dc voltage contribution.

My main issue is the lack of evidence for the role of the SAW in the observed effect, even though the SAW-induced lattice vibrations should be required according to the invoked model. This is a critical point, and the manuscript cannot be published without this being fully resolved.

My critique is as follows:

According to Eq. (5) the acoustic spin rotation is attributed to a spin-orbit field B_{SO} .

- a) The authors write in the main text (page 10, line 174) "The B_{SO} is caused by the SAW-induced z-direction gradient of magnetization,..".
- b) In the SI, the authors write (bottom of page 9). " B_{SO} due to lattice vibration"

Taking point a) and b) together, I understand that B_{SO} is attributed to a z-gradient of magnetization due to the lattice vibration of the SAW. This directly implies that the magnitude of B_{SO} must increase with increasing the SAW amplitude, viz., the microwave power. Without SAW, B_{SO} must vanish.

This is however not in agreement with the experimental observation. Clearly, if B_{SO} is power dependent, then the ratio V_{ASR}/V_{ASP} in Fig. 4d) must depend on power P , which it does not. If there would be any SAW-induced effect (B_{SO} due to SAW), the power dependence in Fig. 4c) would need to be non-linear. Thus, the experimental data contradicts the proposed model.

From the experimental data (Fig. 4d)) one would expect that the same effect can also be observed without any SAW. The authors also write on page 2, line 73 "the spin rotation [...] scales with the interface SO, independent of the magnetic field and SAW". Thus, all experimental data suggests that the SAW plays no role in the observed effect. The critical reader thus asks why did the authors use SAW at all and why should the effect be called "acoustic" spin rotation?

Clearly, a control experiment to demonstrate that the effect is absent without any SAW is required if the authors want to invoke a model of spin rotation based on lattice vibration. A possible control experiment could be the excitation of the spin precession by other means, such as direct microwave excitation combined with application of the SAW.

Additionally, the model needs to be explained better by addressing the following points:

- 1) Why should the SAW cause a gradient of magnetization (line 174)? This needs to be explained in a microscopic picture as it is not clear at all.
- 2) The authors switch from the field B_{SO} to a torque τ in the text (page 10 line 176) and invoke the spin Hall angle for the torque. No explanation or reference is given why this is an

appropriate model. In the SI, neither τ nor the spin Hall angle appear in the expression for B_{SO} , so it remains ominous. The authors need to clearly state the relation of B_{SO} and τ and explain why the interfacial torque τ is proportional to the spin Hall angle.

There are several further issues that should be addressed:

- 3) The authors refer to a phase difference between ASP and ASR voltage in line 62. Both voltages are DC, so there cannot be a phase difference.
- 4) In the next line, they say that this so-called phase difference (presumably they mean difference in angular dependency) is "direct evidence" for "acoustic spin rotation". It is not direct evidence for spin rotation. Direct evidence would be a detection of the spin rotation depicted in Fig. 1. Any measurement of dc voltages is indirect.
- 5) Potential impacts of rectification voltages due to microwave current in the bilayers are not considered. A control experiment with an insulating barrier between the FM and the HM is required to show that both V_{ASP} and V_{ASR} vanish.
- 6) Because the V_{ASR} is attributed to an interface effect (page 10, line 177: "The interface torque arises due to HM/FM interlayer diffusion [...]"), it should vanish in a Pt/Cu/Ni trilayer, where V_{ASP} would be still observed. Such a control experiment is needed.
- 7) The lineshape of the raw V_{xx} vs H data is not convincing. From Fig. 3d) it is not possible to see that the lineshape is a symmetric Lorentzian. Typically, the spin pumping data is fitted by a superposition of a symmetric and antisymmetric Lorentzian to disentangle spin pumping from rectification voltages. Such an analysis is lacking here, there is no information for the reader if the lineshape changes with angle ϕ .
- 8) Why did the authors choose such a low SAW frequency? With a higher frequency such as used in most of the SAW references, it would be much easier to disentangle switching effects from the resonant microwave driving.
- 9) Comparison of microwave absorption in Fig. 2d) and dc voltage in Fig. 3a) is difficult because of the use of different x-scales and only showing 10mT data in Fig. 2d. The same fields and orientations should be shown in both plots.
- 10) The resonance field is 4mT according to the authors (page 5, line 95). Data at 4mT should thus be shown in Fig. 2d) and 3a)

Reviewer #3:

Remarks to the Author:

The authors report significant asymmetric behavior with respect to $\phi = 90^\circ$ in the angular-dependent acoustic spin pumping measurements of V_{xx} for Pt/Ni bilayer. They attribute this observation to acoustic spin rotation, which originates from the interface spin-orbit coupling mediated coupling of electron spins and the lattice. The work is interesting as it may reveal a new coupling between spin current and lattice vibration mediated by spin-orbit coupling. The interpretation is, however, incomplete, and a more detailed analysis seems required before publication. I have several issues with both the measurements and interpretation below.

1. It looks like there are two types of electron spins. One cannot be rotated and contributes to the VASP, while the other is rotated 90 degrees and contributes to the VASR. The author should explain the coexistence of these two electron spins in a uniform system. One would expect all electron spins to exhibit the same behavior, whether they rotate or not. It is also hard to understand why the one kind of spin is rotated by exactly 90 degrees, but not other angles. I recommend that the authors discuss these issues in a revised version.
2. Is the acoustic spin rotation a universal phenomenon on the HM/FM interface, or it relies on the FM materials used for the HM/FM interfaces? In other words, what is reason why other papers did not observe this effect? Is Ni important for this observation? What are the important physical parameters that determine this effect.
3. In order for the reader to better understand the measured signal, it would be better to show the image of Hall device. It could also be interesting to measure the voltage along the y direction and compare it with V_{xx} . One would expect that both signals can reflect the acoustic spin rotation. I recommend that the authors discuss these issues in a revised version.

4. "Under mirror reflection in the xz plane" in Line 101, is it xz plane or yz plane according to Fig. 2a? It does not really make sense to me that the mirror plane is xz plane and the in-plane magnetization m is changed from φ to $180-\varphi$ (or I have missed a central point).
5. It is not easy to follow the interpretation of acoustic spin rotation in the Discussion part. It would be helpful to plot a schematic diagram to show the physical origin of spin rotation in the manuscript.

Response to the reviewers

We thank all the reviewers for their careful reading of the manuscript and their insightful comments. We have incorporated our responses into the revised manuscript as appropriate. Below please find our point-by-point response to the detailed comments of the reviewers.

Questions of the First Reviewer

Q1. The authors claim that the magnetization gradient in the z-direction induced by SAWs produces an effective magnetic field \mathbf{B}_{SO} . If this is true, to clarify the origin of \mathbf{B}_{SO} , the authors should examine the dependence of the acoustic spin rotation signal on both the thickness of the FM layer and the frequency of the SAWs. The SAW attenuation depth along the z-direction decreases with increasing the frequency. Namely, the strength of \mathbf{B}_{SO} should be increased with the frequency.

Re: We gratefully thank the reviewer for the instructive suggestions. Based on the reviewer's suggestion, the surface acoustic wave (SAW) attenuation depth along the z-direction decreases with increasing the frequency. Namely, the strength of \mathbf{B}_{SO} should be increased with the frequency, which leads to the increase of acoustic spin rotation efficiency. Therefore, we should measure the dependence of acoustic spin rotation efficiency on the SAW frequency to prove the accuracy of the acoustic spin rotation model. We set the finger widths and spacings of interdigital transducers to 4.25, 2.87, 2.13, 1.88 and 1.43 μm to obtain the different SAW frequencies, as shown in Fig. R1. The transmission (S_{21}) spectra of SAW devices with different finger widths are shown in Fig. R2. The SAW frequencies are 1.06, 1.70, 2.29, 2.60 and 2.05 GHz, where 2.05

GHz is the third harmonic and the other frequencies are the fifth harmonic. The obtained frequencies are consistent with our device design.

Figure R1. (a-e) Optical microscopy image of SAW devices with finger widths of 4.25, 2.87, 2.13, 1.88 and 1.43 μm , respectively. The dark point is due to the wire bonding.

Figure R2. (a-e) The transmission (S_{21}) spectra of SAW devices with finger widths of 4.25, 2.87, 2.13, 1.88 and 1.43 μm , respectively.

We measured the longitudinal inverse spin Hall effect (ISHE) voltage V_{xx} of Pt(2)/Ni(30) embedded in SAW delay lines with different SAW frequencies. Similar with the discussion of the manuscript, we decompose the asymmetric V_{xx} into two

components that are symmetric V_{xx}^S [Fig. R3(a)] and antisymmetric V_{xx}^A [Fig. R3(b)] with respect to $\varphi = 90^\circ$. For the convenience of comparing the acoustic spin rotation efficiency, we normalize V_{xx}^S to $[-1, 1]$. One can find V_{xx}^S for various frequencies all exhibit the $\sin\varphi\sin^22\varphi$ angular dependences, suggesting the acoustic spin pumping. While V_{xx}^A for various frequencies all exhibit the $\cos\varphi\sin^22\varphi$ angular dependences, which we attribute to the acoustic spin rotation. We can also find that in Fig. R3(b) the acoustic spin rotation voltage increases with the SAW frequency, suggesting acoustic spin rotation efficiency $\eta = V_{ASR}/V_{ASP}$ increases with SAW frequency. As shown in Fig. R3(c), η seems to be proportional to the SAW frequency, which we will discuss as following. Due to the SAW attenuation in the z -direction, the equivalent magnetization during magnetization precession also decays in the form of $\mathbf{M} = \mathbf{M}_0 e^{-z/\lambda}$, where λ is the attenuation depth of SAW. Since λ is inversely proportional to SAW frequency f [Sci. Adv. **7**, eabd9697 (2021)], the gradient of magnetization $\partial\mathbf{M}/\partial z = (\mathbf{M}_0/\lambda)e^{-z/\lambda} \approx \mathbf{M}_0/\lambda \propto f$. Here we consider $e^{-z/\lambda} \approx 1$, because the thickness of the ferromagnetic layer z is in nanometer scale, while λ in our case is about several μm . Therefore, the experimental result of acoustic spin rotation efficiency increasing with SAW frequency in Fig. R3(c) obeys with the theoretical expectation.

Figure R3. (a) The symmetric and (b) the antisymmetric components of V_{xx} of Pt(2)/Ni(30) embedded in SAW delay lines with different SAW frequencies. (c) SAW frequency dependence of acoustic spin rotation efficiency η .

Following the reviewer's another suggestion, we further studied the dependence of acoustic spin rotation efficiency on the Ni thickness d . We measured V_{xx} of Pt/Ni bilayers with different Ni thicknesses at a fixed SAW frequency $f = 2.1$ GHz. In order to clearly show the acoustic spin rotation effect, we decompose V_{xx} of different Ni thicknesses into symmetric component V_{xx}^S and antisymmetric component V_{xx}^A . For the convenience of comparing the acoustic spin rotation efficiency, we also normalize V_{xx}^S of different Ni thicknesses. V_{xx}^S for different Ni thicknesses still satisfy the $\sin\varphi\sin^22\varphi$ angular dependences [Fig. R4(a)]. While V_{xx}^A for different Ni thicknesses all satisfy the $\cos\varphi\sin^22\varphi$ angular dependences [Fig. R4(b)]. However, we note that the antisymmetric acoustic spin rotation voltage increases with increasing Ni thickness. Figure R4(c) shows the acoustic spin rotation efficiency $\eta = V_{ASR}/V_{ASP}$ as a function of Ni thickness. As Ni thickness increases, η increases sharply at beginning, and then tends to saturation after 30 nm. We find that these data can be described by the drift-

diffusion model $\eta(d) = \eta(\infty)(1 - \text{sech}(d/\lambda_{sd}))$ [Nat. Mater. **22**, 591 (2023)], where λ_{sd} is the spin diffusion length. By fitting, $\lambda_{sd} = 13.5$ nm, suggesting the range of acoustic spin rotation in ferromagnetic layer.

We have added the SAW frequency dependence of acoustic spin rotation effect in the revised manuscript (lines 220-228) and the Ni thickness dependence of acoustic spin rotation effect in Section F of the Supplemental Material.

Figure R4. (a) The symmetric and (b) the antisymmetric components of V_{xx} of Pt/Ni bilayers with different Ni thicknesses. (c) Ni thickness dependence of acoustic spin rotation efficiency η .

Q2. The crystal cut surface of the lithium niobate (LN) substrate used in this study must be specific. Furthermore, the authors should describe which mode of SAWs is excited and which strain tensor components contribute to the magnetic dynamics. For instance, in the case of a 128-degree Y-cut LN substrate, it is known that the phase of the field angle variation of the inverse spin Hall voltage possibly depends on the crystalline orientation of the SAW propagation.

Re: We thank the reviewer very much for pointing out this problem, which is important to understanding the spin rotation due to the lattice movement. In our study, we use the 128° Y-cut LiNbO₃ substrate to excite SAW. “The propagation direction of SAW is along the X-crystalline axis of the LiNbO₃ substrate, which is defined as x -axis. Along this propagation direction, the SAW is Rayleigh-mode, which means that the vibration is only along the x and z directions. Therefore, there are three nonvanishing strain components at the surface of the substrate, namely, ϵ_{xx} , ϵ_{zz} , and ϵ_{xz} .” In our acoustic device, the observed $\sin^2 2\varphi$ angular dependence in Fig. 2(d) is a hallmark of acoustic ferromagnetic resonance, strongly suggesting that ϵ_{xx} dominates the magnetization dynamics [Phys. Rev. B **86**, 134415 (2012)]. In addition, ϵ_{xz} also plays an important role, which leads to nonreciprocity as reviewer referred.

Please kindly check it in the revised manuscript (lines 87-92).

Q3. Does the dimension of the formula, which is given in the present paper for the effective interfacial torque due to the effective magnetic field, is the torque per unit area? I believe that the time derivative of magnetization should be included in the formula for torque. If so, the effective magnetic field should depend on frequency. Demonstrating the frequency dependence of the acoustic spin rotation effect is, therefore, very important for understanding the physics.

Re: We appreciate the reviewer’s very good suggestion. According to Zhang's theoretical work, a new torque (also called Zhang-Li torque) is induced by the nonuniform distribution of magnetization [Phys. Rev. Lett. **93**, 127204 (2004)], which is also verified by experiments [Nat. Mater. **16**, 712 (2017); Phys. Rev. B **102**, 214408 (2020); Sci. Rep. **9**, 9592 (2019)]. Owing to the Zhang-Li torque, the SAW attenuation

in the z -direction induces an interfacial torque $\boldsymbol{\tau} = -\frac{c}{M_s} \mathbf{M} \times \partial \mathbf{M} / \partial z$. The LLG equation including the effective interfacial torque is:

$$\frac{\partial \mathbf{M}}{\partial t} = -\gamma \mathbf{M} \times \mu_0 \mathbf{H}_{\text{eff}} + \frac{\alpha}{M_s} \mathbf{M} \times \frac{\partial \mathbf{M}}{\partial t} - \frac{c}{M_s} \mathbf{M} \times \frac{\partial \mathbf{M}}{\partial z}, \quad (\text{R1})$$

where the coefficient c depends on effective interfacial spin Hall angle $\theta_{\text{SH}}^{\text{eff}}$, and has the unit of velocity. Therefore, the unit of the effective interfacial torque in Eq. (R1) should be the torque per unit volume.

The reviewer's another suggestion that the effective interfacial torque should include the time derivative of magnetization $\partial \mathbf{M} / \partial t$ is exactly correct. The nonequilibrium spin density $\delta \mathbf{m}$ that caused interfacial spin-orbit torque $\boldsymbol{\tau} \propto \mathbf{M} \times \delta \mathbf{m}$ is created by two source terms: one is the time variation of the magnetization $\partial \mathbf{M} / \partial t$ and the other is the spatial variation of the magnetization $\nabla \mathbf{M}$. Fortunately, the effective interfacial torque including $\partial \mathbf{M} / \partial t$ can be absorbed in LLG equation by the redefinition of the effective gyromagnetic ratio and the effective Gilbert damping constant. Therefore, in our work we only need to discuss the interfacial torque caused by the spatial variation of the magnetization.

Q4. It should be explicitly commented how the discovery of the acoustic spin rotation effect contributes to device applications. This is crucial for assessing the level of acceptance for publication in Nature Communications.

Re: We sincerely thank the reviewer for the valuable suggestion. Controlling spin direction is the key for device applications, such as for the spin-orbit torque (SOT) device [Nature **476**, 189 (2011); Science **336**, 555 (2012)] and the quantum computing device [Science **346**, 207 (2014)]. For the SOT device, spin direction determines the magnetization dynamics. As a result, controlling spin direction, i.e., spin rotation, has

played an important role in SOT devices with perpendicular magnetic anisotropy, as it induces efficient and field-free switching [Nat. Phys. **13**, 300 (2017); Nat. Commun. **8**, 911 (2017); Phys. Rev. Lett. **121**, 136805 (2018); Nat. Mater. **17**, 509 (2018); Nat. Mater. **19**, 292 (2020); Nat. Commun. **11**, 4671 (2020); Nat. Nanotechnol. **16**, 277 (2021); Nat. Mater. **20**, 800 (2021); Nat. Commun. **12**, 6524 (2021); Nat. Commun. **13**, 4447 (2022)]. On the other hand, spin rotation is also promising to realize quantum computing [Science **346**, 207 (2014), Nature **601**, 338 (2022), Nature **601**, 343 (2022), Nature **601**, 348 (2022), Nat. Commun. **13**, 206 (2022)]. To rotate the spin state from one state to another can be considered as single qubit gate operations in quantum computing, such as Hadamard gate and quantum non gate.

From the application view, the acoustic spin rotation also may help advance the application of novel acoustic spin devices in SOT device and quantum computing. Firstly, the acoustic spin rotation is a conceptual breakthrough, which suggests a new coupling between spin and lattice. “Distinct from the existing electric spin devices that use charge motion to control spin state, for the first time we propose to use lattice motion as a new degree of freedom to realize spin rotation. Secondly, the spin rotation by lattice has been confirmed to be much more efficient than that by the spin rotation in conventional spintronic devices [Nat. Mater. **17**, 509 (2018), Nat. Phys. **18**, 800 (2022)], which can further reduce power consumption. Thirdly, owing to the wave property that enables device to realize noncontact control spin polarization, the acoustic spin rotation will invigorate research activities towards exploring new generation acoustic spintronic devices. This also indicates that we can noncontactly control the magnetic bits in spintronic devices and quantum bits in quantum computing. Finally,

since spin rotation by lattice can realize rotating any in-plane spin state in our work, this indicates that it can realize all single qubit gates for quantum computing device.”

Please kindly check it in the revised manuscript (lines 53-57).

Q5. What intensity of microwaves was used in the experiment in Fig. 5? Comparing the results in Fig. 4 and Fig. 5(f), both the acoustic spin pumping and acoustic spin rotation signals are enhanced when SAWs propagate in the negative-x direction. Is this effect quantitatively explained by the non-reciprocal propagation characteristics of surface acoustic waves in Ni thin films?

Re: We appreciate the reviewer’s very good suggestion. The intensity of microwaves used in the experiment in Fig. 5 is 10 mW. “Comparing the results in Fig. 4 (+x propagation) and Fig. 5(f) (-x propagation), for the same SAW power, both V_{ASP} and V_{ASR} are enhanced when SAW propagates in the -x direction in Pt/Ni bilayer. We attribute it to both the nonreciprocity of system and the difference between two interdigital transducers.”

Please kindly check it in the revised manuscript (lines 189-191).

Questions of the Second Reviewer

Q1. According to Eq. (5) the acoustic spin rotation is attributed to a spin-orbit field \mathbf{B}_{SO} .

a) The authors write in the main text (page 10, line 174) “The \mathbf{B}_{SO} is caused by the SAW-induced z-direction gradient of magnetization,..”.

b) In the SI, the authors write (bottom of page 9). “ \mathbf{B}_{SO} due to lattice vibration”.

Taking point a) and b) together, I understand that \mathbf{B}_{SO} is attributed to a z-gradient of magnetization due to the lattice vibration of the SAW. This directly implies that the magnitude of \mathbf{B}_{SO} must increase with increasing the SAW amplitude, viz., the microwave power. Without SAW, \mathbf{B}_{SO} must vanish.

This is however not in agreement with the experimental observation. Clearly, if \mathbf{B}_{SO} is power dependent, then the ratio $V_{\text{ASR}}/V_{\text{ASP}}$ in Fig. 4(d) must depend on power P , which it does not. If there would be any SAW-induced effect (\mathbf{B}_{SO} due to SAW), the power dependence in Fig. 4(c) would need to be non-linear. Thus, the experimental data contradicts the proposed model.

From the experimental data (Fig. 4(d)) one would expect that the same effect can also be observed without any SAW. The authors also write on page 2, line 73 “the spin rotation [...] scales with the interface SO, independent of the magnetic field and SAW”. Thus, all experimental data suggests that the SAW plays no role in the observed effect. The critical reader thus asks why did the authors use SAW at all and why should the effect be called “acoustic” spin rotation?

Re: We appreciate the reviewer’s very nice comment. The reason that the reviewer’s conclusion cannot match the experimental results is because the reviewer’s assumption that the spin-orbit field \mathbf{B}_{SO} increases with increasing the SAW power is not available. In fact, if reviewer reconsiders that \mathbf{B}_{SO} is independent of SAW power, then the conclusion will be consistent with our experimental results. Now we discuss why \mathbf{B}_{SO} is independent of SAW power.

In our case, the acoustic wave is a Rayleigh surface wave. The lattice displacement in three directions decays along the z-direction, which can be simply described by

following equations: $u_x = u_{x0}e^{-z/\lambda}$, $u_y = 0$ and $u_z = u_{z0}e^{-z/\lambda}$, where λ is the attenuation depth of SAW, as shown in Fig. R5(a). When the SAW acts on the ferromagnet due to magnetoelastic effect, the magnetization precession should also decay along the z -direction, as a result, the time-averaged equivalent magnetization during precession \mathbf{M} can be considered as $\mathbf{M} = \mathbf{M}_0e^{-z/\lambda}$. In our acoustic spin rotation model, \mathbf{B}_{SO} is induced by the gradient of magnetization caused by SAW attenuation. This means $\mathbf{B}_{\text{SO}} \propto \partial\mathbf{M}/\partial z \propto (\mathbf{M}_0/\lambda)e^{-z/\lambda} \approx \mathbf{M}_0/\lambda \propto f$. Here we consider $e^{-z/\lambda} \approx 1$, because the thickness of the ferromagnetic layer z is in nanometer scale, while λ in our case is about several μm . Because the attenuation depth λ is dependent of frequency rather than the power, one can find that $\partial\mathbf{M}/\partial z$ is insensitive to the SAW power, but is proportional to frequency. The corresponding gradient of magnetization $\partial\mathbf{M}/\partial z$ due to the z -direction SAW attenuation is also calculated and shown in Fig. R5(b). Moreover, we also show \mathbf{B}_{SO} is proportional to frequency in Fig. R5(c), by fabricated SAW devices with different finger widths. These results strongly suggest that the experimental data is consistent with the proposed model.

Figure R5. (a) Displacement of Rayleigh wave and (b) magnetization gradient induced by SAW attenuation as a function of z . (c) Frequency dependence of acoustic spin rotation efficiency $\eta \propto \mathbf{B}_{\text{SO}}$.

Q2. Clearly, a control experiment to demonstrate that the effect is absent without any SAW is required if the authors want to invoke a model of spin rotation based on lattice vibration. A possible control experiment could be the excitation of the spin precession by other means, such as direct microwave excitation combined with application of the SAW.

Re: We appreciate the reviewer's very good suggestion. In fact, we always try our best to exclude microwave signal into our sample, by time gating technology and impedance match. This is because when we apply both microwave and SAW on our sample, the rectification signal due to microwave current and electromagnetic induction signal is dominant, which is much larger than the spin pumping signals caused by SAW. The extremely large additional signal makes us very difficult to obtain the spin rotation effect induced by SAW, because of the very low signal-noise ratio. However, we measured only microwave-induced spin pumping signal, by covering a CPW board on our sample. In this case, we do not observe the spin rotation effect when SAW is absent.

Q3. Why should the SAW cause a gradient of magnetization (line 174)? This needs to be explained in a microscopic picture as it is not clear at all.

Re: We appreciate the reviewer's very good suggestion. The microscopic picture of acoustic spin rotation is shown in Fig. R6. "Since in our case the acoustic wave is a Rayleigh surface wave, the lattice displacements decay exponentially along the z -direction. The lattice displacement \mathbf{u} can be simply described by following equations: $u_x = u_{x0}e^{-z/\lambda}$, $u_y = 0$ and $u_z = u_{z0}e^{-z/\lambda}$, where λ is the attenuation depth of

SAW. When SAW acts on the ferromagnet due to magnetoelastic coupling, the dynamic strain due to lattice displacement causes the magnetization precession. As shown in Fig. R6, The z -direction SAW attenuation will lead to the decay of the magnetization precession, as a result, the time-averaged equivalent magnetization \mathbf{M} also decays along the z -direction, which causes a gradient of magnetization along the z -direction.”

We have added the microscopic picture of acoustic spin rotation and explained why the SAW cause a gradient of magnetization in the revised manuscript (lines 199-205).

Figure R6. Schematic diagram of magnetization gradient caused by SAW attenuation.

Q4. The authors switch from the field \mathbf{B}_{SO} to a torque τ in the text (page 10 line 176) and invoke the spin Hall angle for the torque. No explanation or reference is given why this is an appropriate model. In the SI, neither τ nor the spin Hall angle appear in the expression for \mathbf{B}_{SO} , so it remains ominous. The authors need to clearly state the relation of \mathbf{B}_{SO} and τ and explain why the interfacial torque τ is proportional to the spin Hall angle.

Re: We appreciate the reviewer's very good suggestion. The model we used is the Zhang-Li model [Phys. Rev. Lett. **93**, 127204 (2004)]. "According to Zhang's theoretical work, a new torque (also called Zhang-Li torque) is induced by the nonuniform distribution of magnetization [Phys. Rev. Lett. **93**, 127204 (2004)], which is also verified by experiments [Nat. Mater. **16**, 712 (2017); Phys. Rev. B **102**, 214408 (2020); Sci. Rep. **9**, 9592 (2019)]. In our case, nonuniform magnetization $\partial\mathbf{M}/\partial z$ is caused by SAW attenuation, which has been explained in Q3. Owing to the Zhang-Li torque, the SAW attenuation in the z -direction induces an interfacial torque $\boldsymbol{\tau} = -\frac{c}{M_s}\mathbf{M} \times \partial\mathbf{M}/\partial z$. Considering $\boldsymbol{\tau} = \mathbf{M} \times \mathbf{B}_{\text{SO}}$, we have the z -direction spin-orbit field $\mathbf{B}_{\text{SO}} = -\frac{c}{M_s}\partial\mathbf{M}/\partial z$. The coefficient c depends on effective spin Hall angle $\theta_{\text{SH}}^{\text{eff}}$. $\theta_{\text{SH}}^{\text{eff}}$ is proportional to the intrinsic θ_{SH} . The torque arises due to heavy metal (HM)/ferromagnet (FM) interlayer diffusion of a spin accumulation generated in the FM layer, which creates an important source of spin rotation."

Please kindly check it in the revised manuscript (lines 206-214).

Q5. The authors refer to a phase difference between acoustic spin pumping and acoustic spin rotation voltage in line 62. Both voltages are DC, so there cannot be a phase difference.

Re: We appreciate the reviewer's good suggestion. Here the phase difference we referred to is difference in angular dependency of the acoustic spin pumping and the acoustic spin rotation voltages. We will demonstrate the difference in angular dependency in Fig. R7. The angular dependence of V_{xx} for Pt/Ni bilayer in Fig. R7(a) exhibits significant asymmetric behavior with respect to $\varphi = 90^\circ$. According to the

mirror symmetric analysis, we decompose V_{xx} into two components that are symmetric V_{xx}^S [the red line in Fig. R7(b)] and antisymmetric V_{xx}^A [the blue line in Fig. R7(c)] with respect to $\varphi = 90^\circ$. We can easily see that the red line is the almost same with the blue one, when we shift the red line 90° phase to the left. We attribute such 90° difference in angular dependency to the 90° spin rotation during the spin pumping.

Figure R7. (a) The angular dependence of V_{xx} for Pt(2)/Ni(30). We extract V_{xx} into (b) the symmetric and (c) the antisymmetric components with respect to $\varphi = 90^\circ$.

Moreover, we revised all the “phase difference” to the “difference in angular dependency” in the revised manuscript.

Q6. In the next line, they say that this so-called phase difference (presumably they mean difference in angular dependency) is “direct evidence” for “acoustic spin rotation”. It is not direct evidence for spin rotation. Direct evidence would be a detection of the spin rotation depicted in Fig. 1. Any measurement of dc voltages is indirect.

Re: Reviewer is right. It is very difficult for us to provide direct evidence to detect the spin rotation. The acoustic spin rotation effect is obtained from the angular dependence

of the measured dc voltages. Below we will explain why the angular dependence of the measured dc voltages can be considered as evidence of the spin rotation.

Firstly, acoustic spin rotation voltage and acoustic spin pumping voltage exhibit all the same magnetic field dependence, SAW frequency dependence, and SAW power dependence. This indicates that they share the same source, both coming from the pumping spin current.

Secondly, the longitudinal acoustic spin pumping voltage $V_{xx}^{ASP} \propto \sin\varphi \sin^2 2\varphi$ while the longitudinal acoustic spin rotation voltage $V_{xx}^{ASR} \propto \cos\varphi \sin^2 2\varphi$. Since acoustic ferromagnetic resonance exhibit $\sin^2 2\varphi$ angular dependence [Fig. 2(d)], the amplitude of injected spin current also has $\sin^2 2\varphi$ angular dependence, and this enters $V_{xx}^{ASP}, V_{xx}^{ASR} \propto \sin^2 2\varphi$. The rest contribution of the $\sin\varphi$ ($\cos\varphi$) dependence is due to the projection of V_{ASP} (V_{ASR}) along the x -direction, as shown in Fig. R8. This agrees well with the picture of the 90° spin rotation. When V_{ASP} is rotated 90° to V_{ASR} , the projection along the x -direction changes from $\sin\varphi$ to $\cos\varphi$.

Figure R8. Schematic diagram of σ (σ') induced ISHE voltage V_{ASP} (V_{ASR}).

Thirdly, we also measured the transverse ISHE voltage V_{xy} of the Hall bar device, which can also reflect acoustic spin rotation in HM/FM bilayer. We experimentally obtain that the transverse acoustic spin pumping voltage $V_{xy}^{ASP} \propto -\cos\varphi \sin^2 2\varphi$ while the transverse acoustic spin rotation voltage $V_{xy}^{ASR} \propto \sin\varphi \sin^2 2\varphi$. Similar with the

longitudinal ISHE voltage, $V_{xy}^{ASP}, V_{xy}^{ASR} \propto \sin^2 2\varphi$. The rest contribution of the $-\cos\varphi$ ($\sin\varphi$) dependence is derived from the projection of \mathbf{V}_{ASP} (\mathbf{V}_{ASR}) along the y-direction. This provides strong evidence of spin rotation because when \mathbf{V}_{ASP} is rotated 90° to \mathbf{V}_{ASR} , the projection along the y-direction changes exactly from $-\cos\varphi$ to $\sin\varphi$.

Moreover, the statement of “direct evidence” is not correct and we revised it to the “strong evidence”. Please kindly check it in the revised manuscript (lines 68).

Q7. Potential impacts of rectification voltages due to microwave current in the bilayers are not considered. A control experiment with an insulating barrier between the FM and the HM is required to show that both V_{ASP} and V_{ASR} vanish.

Re: We gratefully thank the reviewer for the valuable suggestion. “We can exclude microwave induced rectification voltage from the following three aspects.

Firstly, the rectification voltage cannot cause such an angular dependence. When the SAW passes through the Ni stripe, the dynamic strains cause Ni stripe deformation, which further drives magnetization oscillation due to the magnetoelastic coupling. Since the resistivity of the Ni stripe depends on the direction of the magnetization, the SAW-induced magnetization oscillation will induce the variation of the resistivity of Ni stripe. When the time-dependent resistivity is coupled to microwave current, a dc rectified voltage will be generated. According to the work of Chen *et al.* [Adv. Mater. **35**, 2302454 (2023)], the angular dependence of the SAW-driven rectification Hall voltage in Ni monolayer is $\sin 2\varphi \sin\varphi$ or $\sin 2\varphi$. While the Hall voltage in our work exhibits the $\cos\varphi \sin^2 2\varphi$ and $\sin\varphi \sin^2 2\varphi$ angular dependence.

Secondly, following the reviewer's suggestion, we design a control experiment by inserting a SiO₂ layer into Pt/Ni bilayer. Figure R9 shows the longitudinal ISHE voltage V_{xx} of Pt(2)/Ni(30) and Pt(2)/SiO₂(50)/Ni(30) samples. When SiO₂ is inserted between Pt and Ni, V_{ASP} vanishes, demonstrating that the pumping spin current is blocked by the nonmagnetic insulating SiO₂ layer. However, we can note that V_{ASR} also vanishes, suggesting it originates from the pumping spin current, rather than the microwave induced rectification voltage.

Figure R9. The angular dependence of V_{xx} for Pt(2)/Ni(30) and Pt(2)/SiO₂(50)/Ni(30) samples.

Thirdly, we have carefully filtered out the influence of microwaves in our SAW devices, by optimizing the design to match the impedance. Figure R10 shows the time-domain results of our SAW device measured by time-resolved methods [Appl. Phys. Lett. **119**, 012401 (2021)]. The red line in Fig. R10 is the pulse signal input to interdigital transducer 1, while the blue line is the output signal on interdigital transducer 2. Because the traveling speed of SAW is only 3800 m/s, which is much lower than 3×10^8 m/s of the electromagnetic wave, one can find that SAW-induced signal has a strong delay, but electromagnetic wave-induced signal almost occurs at the same position as the input signal (in the red dotted frame). The almost clean signal in

the red dotted frame suggests the influence of microwaves is negligible in our SAW devices.

Figure R10. The time-domain measurement results of the SAW device. The input signal (red line) was applied on interdigital transducer 1; the response signal (blue line) was detected on interdigital transducer 2.”

Please kindly check it in the revised manuscript (lines 134-138) and Section G of the Supplemental Material.

Q8. Because the V_{ASR} is attributed to an interface effect (page 10, line 177: “The interface torque arises due to HM/FM interlayer diffusion [...]”), it should vanish in a Pt/Cu/Ni trilayer, where V_{ASP} would be still observed. Such a control experiment is needed.

Re: We sincerely thank the reviewer for the valuable suggestion. “In our acoustic spin rotation model, the acoustic spin rotation effect originates from the Pt/Ni interface. To prove this, we inserted a Cu layer between Pt and Ni. Figure R11(a) shows the longitudinal ISHE voltage V_{xx} of Pt(2)/Ni(30), Pt(2)/Cu(6)/Ni(30), and Pt(2)/Cu(50)/Ni(30) samples. When inserting 6 nm Cu between Pt and Ni, the

symmetric V_{ASP} was reduced by approximately 40% due to the diffusion of spin current in Cu [see Fig. R11(b)]. In contrast, the antisymmetric V_{ASR} decreases more (70%) [see Fig. R11(c)], resulting in a 50% reduction of acoustic spin rotation efficiency η . This indicates that the acoustic spin rotation effect originates from the Pt/Ni interface, because the Cu insertion layer destroys the interface between Pt and Ni, leading to a decrease of \mathbf{B}_{SO} . When a thicker Cu (50 nm) is inserted, the spin current vanishes due to the spin relaxation in Cu. As a result, the acoustic spin pumping voltage and acoustic spin rotation voltage both vanish, as shown as the blue scatters in Fig. R11(a).

Figure R11. (a) The angular dependence of V_{xx} for Pt(2)/Ni(30), Pt(2)/Cu(6)/Ni(30), and Pt(2)/Cu(50)/Ni(30) samples. We extract V_{xx} into (b) the symmetric and (c) the symmetric components with respect to $\varphi = 90^\circ$.

Please kindly check it in Section H of the Supplemental Material.

Q9. The lineshape of the raw V_{xx} vs H data is not convincing. From Fig. 3(d) it is not possible to see that the lineshape is a symmetric Lorentzian. Typically, the spin pumping data is fitted by a superposition of a symmetric and antisymmetric Lorentzian to disentangle spin pumping from rectification voltages. Such an analysis is lacking here, there is no information for the reader if the lineshape changes with angle φ .

Re: We appreciate the reviewer's very good suggestion. Figure 3 in the manuscript is not raw V_{xx} , but the fitting coefficient V_{ASP} as a function of H . V_{ASP} is proportional to $\sin\varphi\sin^22\varphi$, suggesting it originates from the acoustic spin pumping. Therefore, V_{ASP} should be a symmetric Lorentz lineshape. Following the suggestion of the reviewer, the microwave-induced rectification voltage that may exhibit antisymmetric Lorentz lineshape [Phys. Rep. **661**, 1 (2016)] is possible mixed in our signal. "We also provide the raw V_{xx} as a function of H , as shown in Fig. R12. Because acoustic ferromagnetic resonance is strongest near $\varphi = k\pi/2 + \pi/4$ ($k \in \text{integer}$) [Fig. 2(d)], we only display raw V_{xx} at -45° , 45° , 135° and 225° . We decompose raw V_{xx} into symmetric [V_S in Fig. R12(b)] and antisymmetric [V_A in Fig. R12(c)] Lorentz lineshape, according to the method in Ref. [Adv. Mater. **35**, 2302454 (2023)]. One can find that the antisymmetric signals for all angles are negligible compared to the symmetric signals. This suggests the microwave-induced rectification voltage is negligible in our system. Therefore, the raw V_{xx} is only a symmetric Lorentz lineshape. Besides, we can observe a significant asymmetry of V_S with respect to $\varphi = 90^\circ$. V_S is larger at 45° (225°) than that at 135° (-45°). We attribute the asymmetry to the acoustic spin rotation.

Figure R12. (a) Raw V_{xx} as a function of H at $\varphi = -45^\circ, 45^\circ, 135^\circ$ and 225° . (b) Symmetric (V_S) and (c) antisymmetric (V_A) signals of V_{xx} .

Please kindly check it in the revised manuscript (lines 147-150) and Section I of the Supplemental Material.

Q10. Why did the authors choose such a low SAW frequency? With a higher frequency such as used in most of the SAW references, it would be much easier to disentangle switching effects from the resonant microwave driving.

Re: We appreciate the reviewer's very good suggestion. The frequency of our SAW device is limited by our lithography technology. The current lithography limit is $1.4 \mu\text{m}$, and the maximum SAW fundamental frequency is 0.68 GHz , which is still lower than the resonant frequency of ferromagnetic resonance. Since we do not have electron beam lithography to pattern the narrower finger width, in our work we use the high-order harmonics of SAW to drive acoustic ferromagnetic resonance.

In order to check that the ferromagnetic resonance is caused by SAW rather than microwave, we measure the S_{21} parameter between the two interdigital transducers as a function of magnetic field orientation. As shown in Fig. R13, S_{21} parameter has a

significant $\sin^2 2\varphi$ angular dependence, rather than microwave-induced $\sin^2 \varphi$ or $\cos^2 \varphi$ angular dependence, when the amplitude of magnetic field is close to resonant field of ferromagnetic film. Because the $\sin^2 2\varphi$ angular dependence is the fingerprint of acoustic ferromagnetic resonance [Phys. Rev. Lett. **106**, 117601 (2011)], we believe that the ferromagnetic resonance is caused by SAW, rather than microwave.

Figure R13. The angular dependence of S_{21} for Pt/Ni under the 5th harmonic SAW for $\mu_0 H = 4$ mT.

Q11. Comparison of microwave absorption in Fig. 2(d) and dc voltage in Fig. 3(a) is difficult because of the use of different x-scales and only showing 10 mT data in Fig. 2(d). The same fields and orientations should be shown in both plots.

Re: Reviewer is right. The same fields and orientations should be shown in both plots. We have replotted Fig. 2(d) with the same x-scales and fields as those in Fig. 3(a), as shown in Figs. R14 and R15. “Figure R14 shows that the S_{21} curves for magnetic fields H from 4 to 18 mT all exhibit the $\sin^2 2\varphi$ angular dependence. This is the fingerprint of acoustic ferromagnetic resonance that distinguishes from microwave-field induced ferromagnetic resonance [Phys. Rev. Lett. **106**, 117601 (2011)]. As H is away from the acoustic ferromagnetic resonance, the power absorption S_{21} gradually decreases.” Figure R15 shows the angular dependence of ISHE voltage V_{xx} for

magnetic fields H from 4 to 18 mT. The symmetric components V_{xx}^S [Fig. R15(b)] for different Hall exhibit $\sin\phi\sin^22\phi$ angular dependence, while the antisymmetric components V_{xx}^A [Fig. R15(c)] all exhibit $\cos\phi\sin^22\phi$ angular dependence. The data in Figs. R14 and R15 are still consistent with our model.

Please kindly check it in the revised manuscript (lines 106-110).

Figure R14. The angular dependence of S_{21} for Pt/Ni under the 5th harmonic SAW for different H .

Figure R15. The angular dependence of V_{xx} for Pt/Ni under the 5th harmonic SAW excitation with $P = 10$ mW for different H . We extract V_{xx} into (b) the symmetric and (c) the antisymmetric components with respect to $\phi = 90^\circ$.

Q12. The resonance field is 4 mT according to the authors (page 5, line 95). Data at 4 mT should thus be shown in Figs. 2(d) and 3(a).

Re: Reviewer is right and we appreciate the reviewer's good suggestion. Because 4 mT is close to the coercivity, we are worried that the data at 4 mT is not typical for the coherent rotation model. As shown in Figs. R14 and R15, the angular dependence of S_{21} and V_{xx} at 4 mT exhibit the same behavior with the other magnetic fields, which does not affect the analysis of the origin of V_{xx} . We have added the data at resonance field (4 mT) in Figs. 2(d) and 3(a).

Questions of the Third Reviewer

Q1. It looks like there are two types of electron spins. One cannot be rotated and contributes to the V_{ASP} , while the other is rotated 90 degrees and contributes to the V_{ASR} . The author should explain the coexistence of these two electron spins in a uniform system. One would expect all electron spins to exhibit the same behavior, whether they rotate or not. It is also hard to understand why the one kind of spin is rotated by exactly 90 degrees, but not other angles. I recommend that the authors discuss these issues in a revised version.

Re: We gratefully thank the reviewer for the valuable suggestion. It's not strange to observe two polarized spin currents coexisting in a uniform system. "For example, when applying an electric field to drive electron motion in the FM/HM system, the charge current can be converted into two polarized spin currents due to the spin-orbit coupling. One is the conventional y-polarized spin current. The other is anomalous z-polarized spin current, which has been widely reported [Nat. Phys. **13**, 300 (2017); Nat.

Commun. **8**, 911 (2017); Phys. Rev. Lett. **121**, 136805 (2018); Nat. Mater. **17**, 509 (2018); Nat. Mater. **19**, 292 (2020); Nat. Commun. **11**, 4671 (2020); Nat. Nanotechnol. **16**, 277 (2021); Nat. Mater. **20**, 800 (2021); Nat. Commun. **12**, 6524 (2021); Nat. Commun. **13**, 4447 (2022)]. Amin *et al.* describe such anomalous z -polarized spin current to the spin-orbit precession effect around the y -direction interface Rashba field, as shown in Fig. R16. [Phys. Rev. Lett. **121**, 136805 (2018)]. Due to the spin precession effect, the injected σ polarization spin current can be rotated around y -direction interface Rashba field to σ' polarization spin current, i.e., $\sigma' = \sigma \times \mathbf{y}$, where σ' is the spin direction of spin current under spin precession, \mathbf{y} is the direction of the interface Rashba field, and σ is the spin direction of the injected spin current. For in-plane ferromagnetic magnetization \mathbf{m} , and $\sigma \parallel \mathbf{m}$, the spin-orbit precession will cause a z -polarized spin current. Because in our case the effective spin-orbit field is along z -direction, i.e., $\sigma' = \sigma \times \mathbf{z}$, as a result, when injecting in-plane σ spin current, we will find the σ' due to spin rotation is always exactly 90° ."

Please kindly check it in the revised manuscript (lines 230-243).

Figure R16. Schematics of spin-orbit precession. Spin-orbit precession occurs when electrons precess about the interfacial Rashba field during reflection and transmission. [Phys. Rev. Lett. **121**, 136805 (2018)].

Q2. Is the acoustic spin rotation a universal phenomenon on the HM/FM interface, or it relies on the FM materials used for the HM/FM interfaces? In other words, what is reason why other papers did not observe this effect? Is Ni important for this observation? What are the important physical parameters that determine this effect?

Re: We gratefully thank the reviewer for the instructive suggestion. We believe the acoustic spin rotation is a universal phenomenon on the HM/FM interface. Because in the strong spin-orbit interaction (SOI) system, spin is not a good quantum number, which means spin is no longer conserved in strong SOI system. Recent experiments and theories have demonstrated the spin rotation existing in various SOI systems, such as two-dimensional Weyl semimetals WTe₂ [Nat. Phys. **13**, 300 (2017)] and MoTe₂ [Nat. Mater. **19**, 292 (2020)], antiferromagnets Mn₂Au [Nat. Mater. **20**, 800 (2021)], Mn₃GaN [Nat. Commun. **11**, 4671 (2020)], Mn₃SnN [Nat. Commun. **12**, 6524 (2021)] and Mn₃Sn [Nat. Commun. **13**, 4447 (2022)], as well as single crystal heavy metal CuPt alloy [Nat. Nanotechnol. **16**, 277 (2021)]. However, with the Galilean principle of relativity, a moving itinerant electron in the lattice is equivalent to an electron in the moving lattice frame. This indicates that an electron can feel an equivalent spin-orbit field due to the lattice moving [Nature **117**, 514 (1926)]. In our acoustic spin devices, we utilize lattice motion replacing electron motion to realize spin rotation. The reason we choose Ni is because Ni exhibits strong magnetoelastic coupling. Moreover, the low saturation magnetization of Ni can also lower the resonant frequency. Besides Ni, we believe Co or CoFeB are also good candidates to observe the acoustic spin rotation effect, but exhibit the larger resonant frequency. Finally, we notice that the antisymmetric acoustic voltage is also observed in strong Rashba system, i.e.,

Ni/Cu/Bi₂O₃ and Ni/Ag/Bi₂O₃ hybrid devices [Phys. Rev. B **97**, 180301 (2018)]. However, they did not dig it, but simply attribute it to the SAW attenuation along the propagation direction. We have excluded their explanation by reversing the SAW propagation as shown in Fig. 5(e).

The next question is what are the important physical parameters that determine this effect. According to our model, we believe the most important physical parameters are SAW frequency f and the spin Hall angle θ_{SH} , which we will discuss as following. The acoustic spin rotation in our system originates from the interface SOI-induced spin-orbit field \mathbf{B}_{SO} . We ascribe \mathbf{B}_{SO} to the SAW-induced z -direction gradient of magnetization. According to Zhang's theoretical work, a new torque (also called Zhang-Li torque) is induced by the nonuniform distribution of magnetization [Phys. Rev. Lett. **93**, 127204 (2004)], which is also verified by experiments [Nat. Mater. **16**, 712 (2017); Phys. Rev. B **102**, 214408 (2020); Sci. Rep. **9**, 9592 (2019)]. Owing to the Zhang-Li torque, the SAW attenuation in the z -direction induces an interfacial torque $\boldsymbol{\tau} = -\frac{c}{M_s} \mathbf{M} \times \partial \mathbf{M} / \partial z$ and the corresponding z -direction spin-orbit field $\mathbf{B}_{\text{SO}} = -\frac{c}{M_s} \partial \mathbf{M} / \partial z$, where the coefficient c depends on effective interfacial spin Hall angle $\theta_{\text{SH}}^{\text{eff}}$. Because the magnetization gradient $\partial \mathbf{M} / \partial z$ is proportional to the SAW frequency, we have experimentally and theoretically confirmed that the acoustic spin rotation effect is almost proportional to the SAW frequency. Moreover, \mathbf{B}_{SO} is also proportional to coefficient c , which is related with the spin Hall angle θ_{SH} . Therefore, we can expect the acoustic spin rotation effect to be more significant with a larger θ_{SH} .

Q3. In order for the reader to better understand the measured signal, it would be better to show the image of Hall device. It could also be interesting to measure the voltage

along the y direction and compare it with V_{xx} . One would expect that both signals can reflect the acoustic spin rotation. I recommend that the authors discuss these issues in a revised version.

Re: We gratefully thank the reviewer for the instructive suggestion. Following the suggestion of the reviewer, we add the image of Hall device, as shown in Fig. R17.

Figure R17. Representative optical microscopy image of the device. A Hall bar made of the HM/FM bilayer is placed between the two interdigital transducers.

“We also measured the transverse ISHE voltage V_{xy} of the Hall bar device, which can further prove the acoustic spin rotation model. Figures R18(a) and (b) show the measured longitudinal ISHE voltage V_{xx} and transverse ISHE voltage V_{xy} of Pt/Ni bilayers, respectively. For the convenience of comparing V_{xx} and V_{xy} , we normalize them both to [-1, 1]. Similar with the discussion of the manuscript, we decompose the asymmetric V_{xx} into two components that are symmetric V_{xx}^S [Fig. R18(c)] and antisymmetric V_{xx}^A [Fig. R18(e)] with respect to $\varphi = 90^\circ$. The angular dependence of V_{xx}^S and V_{xx}^A can be fitted very well by following functions

$$V_{xx}^{ASP} = V_{ASP} \sin \varphi \sin^2 2\varphi, \quad (\text{R2})$$

$$V_{xx}^{ASR} = V_{ASR} \cos \varphi \sin^2 2\varphi, \quad (\text{R3})$$

where V_{ASP} and V_{ASR} represent the amplitudes of acoustic spin pumping voltage induced by σ [Phys. Rev. Lett. **108**, 176601 (2012)] and acoustic spin rotation voltage induced by σ' , respectively. Since acoustic ferromagnetic resonance exhibit $\sin^2 2\varphi$

angular dependence [Fig. 2(d)], the amplitude of injected spin current also has $\sin^2 2\varphi$ angular dependence, and this enters $V_{xx}^{ASP}, V_{xx}^{ASR} \propto \sin^2 2\varphi$. The rest contribution of the $\sin\varphi$ ($\cos\varphi$) dependence is due to the projection of σ (σ') induced ISHE voltage along the x -direction, as shown in Fig. R19.

Figure R18. The angular dependence of (a) longitudinal ISHE voltage V_{xx} and (b) transverse ISHE voltage V_{xy} for Pt/Ni bilayers. V_{xx} is extracted into (c) the symmetric and (e) the antisymmetric components with respect to $\varphi = 90^\circ$. V_{xy} is extracted into (d) the antisymmetric and (f) the symmetric components with respect to $\varphi = 90^\circ$.

Figure R19. Schematic diagram of σ (σ') induced ISHE voltage V_{ASP} (V_{ASR}).

When considering the projection of σ and σ' induced ISHE voltage along the y -direction, we can directly write

$$V_{xy}^{ASP} = -V_{ASP} \cos\varphi \sin^2 2\varphi, \quad (R4)$$

$$V_{xy}^{ASR} = V_{ASR} \sin\varphi \sin^2 2\varphi. \quad (R5)$$

According to Eqs. (R4) and (R5), we also decompose V_{xy} into two components that are antisymmetric [Fig. R18(d)] and symmetric [Fig. R18(f)] with respect to $\varphi = 90^\circ$, which can be fitted very well by Eqs. (R4) and (R5), respectively. It should be noted that there is a negative sign difference between V_{xy}^{ASP} in Fig. R18(d) and V_{xx}^{ASR} in Fig. R18(e). This can be easily understood by the following analysis. As shown in Fig. R19, when V_{ASP} is rotated 90° to V_{ASR} , the projection along the x -direction changes from $\sin\varphi$ to $\cos\varphi$, and the projection along the y -direction changes from $-\cos\varphi$ to $\sin\varphi$.

Thus, the angular dependence of the measured Hall voltage is completely consistent with our acoustic spin rotation model.”

Please kindly check it in the revised manuscript (lines 131-134) and Section E of the Supplemental Material.

Q4. “Under mirror reflection in the xz plane” in Line 101, is it xz plane or yz plane according to Fig. 2(a)? It does not really make sense to me that the mirror plane is xz plane and the in-plane magnetization \mathbf{m} is changed from φ to $180^\circ - \varphi$ (or I have missed a central point).

Re: We thank the reviewer for pointing out this question. This is because the magnetic moment \mathbf{m} is an axis vector, also called pseudovector suggesting that it is not a real vector. For example, \mathbf{m} can be considered to be generated by the current coil, so the direction of \mathbf{m} *only* depends on the current coil. As shown in Fig. R20, the current coil perfectly satisfies the mirror reflection, but \mathbf{m} does not, if we consider \mathbf{m} is generated

by the current coil. One can easily find that, under the mirror reflection, the magnetic moment component parallel to the mirror changes direction, while the magnetic moment component perpendicular to the mirror does not change direction. Therefore, under the mirror reflection of the xz plane σ_{xz} discussed in the main text, \mathbf{m} is changed from φ to $180^\circ - \varphi$, as shown in Fig. R21. The same reason applies to magnetic field \mathbf{H} and spin $\boldsymbol{\sigma}$.

Figure R20. A loop of wire (black), carrying a current I , creates a magnetic moment \mathbf{m} (blue). Under the mirror reflection indicated by the dashed line, the magnetic moment parallel to the mirror changes direction, while the magnetic moment perpendicular to the mirror does not change direction.

Figure R21. Schematics of σ_{xz} mirror symmetry of the SAW device.

Q5. It is not easy to follow the interpretation of acoustic spin rotation in the Discussion part. It would be helpful to plot a schematic diagram to show the physical origin of spin rotation in the manuscript.

Re: We apologize for the unclear description in the manuscript. We have reorganized the interpretation of acoustic spin rotation. Figure R22 shows the schematic diagram of the physical origin of acoustic spin rotation. The acoustic spin rotation originates from the SAW attenuation in the z -direction. “Since in our case the acoustic wave is a Rayleigh surface wave, the lattice displacements decay exponentially along the z -direction. When SAW acts on the ferromagnet due to magnetoelastic coupling, the dynamic strain due to lattice displacement causes the magnetization precession. As shown in Fig. R22, the z -direction SAW attenuation will lead to the decay of the magnetization precession, as a result, the time-averaged equivalent magnetization \mathbf{M} also decays along the z -direction, which causes a gradient of magnetization along the z -direction. Therefore, the magnetization is nonuniform along z -direction. According to Zhang's theoretical work, a new torque (also called Zhang-Li torque) is induced by the nonuniform distribution of magnetization [Phys. Rev. Lett. **93**, 127204 (2004)], which is also verified by experiments [Nat. Mater. **16**, 712 (2017); Phys. Rev. B **102**, 214408 (2020); Sci. Rep. **9**, 9592 (2019)]. Owing to the Zhang-Li torque, the SAW attenuation in the z -direction induces an interfacial torque $\boldsymbol{\tau} = -\frac{c}{M_s} \mathbf{M} \times \partial \mathbf{M} / \partial z$. Thus, a z -direction spin-orbit field $\mathbf{B}_{SO} = -\frac{c}{M_s} \partial \mathbf{M} / \partial z$ is produced. The coefficient c depends on effective interfacial spin Hall angle θ_{SH}^{eff} . θ_{SH}^{eff} is proportional to the intrinsic θ_{SH} .” \mathbf{B}_{SO} induces the injected spin $\boldsymbol{\sigma}$ to rotate 90° to $\boldsymbol{\sigma}'$ via the spin precession, namely the acoustic spin rotation.

Please kindly check it in the revised manuscript (lines 199-212).

Figure R22. Schematic diagram of the physical origin of acoustic spin rotation. Due to the gradient of magnetization caused by SAW attenuation, a spin-orbit field $\mathbf{B}_{\text{SO}} = -\frac{c}{M_s} \frac{\partial \mathbf{M}}{\partial z}$ is induced.

Reviewers' Comments:

Reviewer #1:

Remarks to the Author:

In my opinion, the authors have made adequate changes in the manuscript and replied all questions sufficiently. Therefore, I recommend the publication of this manuscript in Nature Communications.

Reviewer #2:

Remarks to the Author:

All reviewers found the study intriguing but raised several points of concern regarding the interpretation of the results. The authors have provided an extensive reply to all review questions, carried out additional experimental evaluation and revised the manuscript in several instances. The supplemental material is now much more extensive and contains valuable additional information. Overall, I believe that the manuscript has been greatly improved and I now believe that there is indeed experimental evidence for a dc voltage contribution that cannot be explained by acoustic spin pumping or rectification.

From the revised manuscript, I understand that the ratio η of acoustic spin rotation voltage to acoustic spin pumping voltage is independent of power because both dc voltages are linear in SAW power. The authors study the frequency dependency of η and find that the acoustic spin rotation becomes more efficient at higher frequencies which they attribute to the reduced acoustic wavelength and concomitant increased gradient in dynamic magnetization. Additionally, the authors also show a dependence of the acoustic spin rotation voltage on Ni thickness (Figure S5). While I agree with the authors that the general trend in frequency and thickness dependency qualitatively supports their interpretation, I am still not fully convinced by the simple fit models used for both and still miss a more quantitative model.

My remaining concerns are:

- 1) The assumption that the gradient in dynamic magnetization strictly follows the strain gradient is not necessarily good. On the one hand, because the spins in Ni are exchange coupled, one would expect that in general the gradient in magnetization is smaller than the gradient in SAW amplitude if there is no pinning. On the other hand, a gradient in dynamic magnetization can be caused even by uniform SAW in the presence of pinning. If the authors assume that the magnetization gradient strictly follows the strain gradient, they should be able to provide a quantitative number for the magnetization gradient and compare this to previous studies that invoked the Zhang-Li torque. Such a quantitative comparison is missing.
- 2) The authors specify the strength of the spin orbit field BSO only in a semi-quantitative manner, with a parameter c that depends on the "interfacial spin Hall angle". This makes it impossible to judge if the strain gradient / magnetization gradient is sufficiently large to cause a BSO that can result in a measurable torque. Here, at least an order-of-magnitude estimate of BSO is required.

In total, the experimental results are intriguing, and the model is in qualitative agreement with the data. From the revised manuscript it is not possible to see if the proposed model would also reproduce the observed order of magnitude of the effect, which is a remaining weak point of the manuscript.

Reviewer #3:

Remarks to the Author:

I think the authors have addressed the questions raised by the referees satisfactorily. Experimental details and the origin of the observed phenomena are well explained. I recommend the publication of the manuscript.

Response to the reviewers

We appreciate all the reviewers for their positive comments and insightful suggestions. We have incorporated our responses into the revised manuscript as appropriate. Below please find our point-by-point response to the detailed comments of the reviewers.

Questions of the First Reviewer

Q1. In my opinion, the authors have made adequate changes in the manuscript and replied all questions sufficiently. Therefore, I recommend the publication of this manuscript in *Nature Communications*.

Re: We sincerely appreciate the reviewer for the recommendation and previous constructive suggestions that greatly enhance the quality of our manuscript.

Questions of the Second Reviewer

Q1. The assumption that the gradient in dynamic magnetization strictly follows the strain gradient is not necessarily good. On the one hand, because the spins in Ni are exchange coupled, one would expect that in general the gradient in magnetization is smaller than the gradient in SAW amplitude if there is no pinning. On the other hand, a gradient in dynamic magnetization can be caused even by uniform SAW in the presence of pinning. If the authors assume that the magnetization gradient strictly follows the strain gradient, they should be able to provide a quantitative number for the magnetization gradient and compare this to previous studies that invoked the Zhang-Li torque. Such a quantitative comparison is missing.

Re: Reviewer is right. It is very important and necessary to provide a quantitative number for the magnetization gradient, and to compare this to previous studies that invoked the Zhang-Li torque. Here we assume that the magnetization gradient is mainly caused by SAW attenuation, and strictly follows the strain gradient. Due to the SAW attenuation in the z -direction, the equivalent magnetization during magnetization precession also decays in the form of $M = M_s e^{-\frac{z}{\lambda}}$. In our case, the saturated magnetization of Ni $M_s = 4.85 \times 10^5$ A/m, and the attenuation depth of SAW λ is close to the wavelength of SAW [Sci. Adv. 7, eabd9697 (2021)], i.e., $\lambda \approx 17$ μm . Therefore, we can calculate the gradient of magnetization induced by SAW

$$\frac{\partial M}{\partial z} = \frac{M_s}{\lambda} e^{-\frac{z}{\lambda}} \approx \frac{M_s}{\lambda} = 2.9 \times 10^{10} \text{ A/m}^2. \quad (\text{R1})$$

Next, we will compare this value to previous studies that invoked the Zhang-Li torque. By growing Pt/Co/Ni/Co/Pt SOT device on PMN-PT piezoelectric substrate, the Zhang-Li torque can also be generated by applying an additional x -direction electric-field on PMN-PT substrate [Nat. Mater. 16, 712 (2017)]. As shown in Fig. R1(a), the external electric-field will lead to electrical potential gradient at PMN-PT/Pt interface, which can be considered as a spin current source due to the spin-orbit coupling at PMN-PT/Pt interface. Because such interfacial electrical potential exhibits x -direction gradient, the generated spin current \mathbf{J}_s correspondingly exhibits x -direction gradient $\frac{\partial \mathbf{J}_s}{\partial x}$, as shown in Fig. R1(b). When the spin current exerts a damping-like torque on perpendicular magnetization of Co/Ni/Co, it will cause the magnetization deflection along the x direction [see Fig. R1(b)]. Therefore, the spin current gradient $\frac{\partial \mathbf{J}_s}{\partial x}$ will generate the magnetization gradient $\frac{\partial \mathbf{M}_x}{\partial x}$, which induces a Zhang-Li torque $\boldsymbol{\tau} = -\frac{c}{M_s} \mathbf{M} \times \frac{\partial \mathbf{M}_x}{\partial x}$. According the macro-spin model in their study, the spin current

induces the magnetization deflection along the x -direction, $M_x \approx \frac{\hbar J_s}{2edB_{an}^0}$, where \hbar is the reduced Planck constant, e is the elementary charge, d is the thickness of Co/Ni/Co (~ 1.0 nm) and B_{an}^0 is the perpendicular anisotropy field (~ 0.1 T). Therefore, the magnetization gradient induced by interfacial electrical field is

$$\frac{\partial M_x}{\partial x} \approx \frac{\hbar}{2edB_{an}^0} \frac{\partial J_s}{\partial x} = 4.3 \times 10^6 \text{ A/m}^2, \quad (\text{R2})$$

where the spin current gradient $\frac{\partial J_s}{\partial x} = 1.3 \times 10^{12} \text{ A/m}^3$. One can easily find the magnetization gradient induced by SAW in Eq. (R1) is four orders of magnitude larger than that induced by interfacial electrical field, which is sufficiently large to cause a \mathbf{B}_{SO} that can result in a measurable torque. If we further consider the exchange coupling or pinning effect, the magnetization will decay more sharply along the z direction, thus will enhance the magnetization gradient.

Figure R1. (a) The electrical potential gradient at PMN-PT/Pt interface induced by the external electric-field due to the polarization of substrate. (b) The spin current gradient generated by the electrical potential gradient due to the spin-orbit coupling at PMN-PT/Pt interface. The total spin current \mathbf{J}_s is the superposition of the two effects - the spin Hall effect and the spin-orbit coupling. [Nat. mater. 16, 712 (2017)]

Please kindly check the quantitative estimation of magnetization gradient in the revised manuscript (lines 220-230, on page 12).

Q2. The authors specify the strength of the spin orbit field \mathbf{B}_{SO} only in a semi-quantitative manner, with a parameter c that depends on the “interfacial spin Hall angle”. This makes it impossible to judge if the strain gradient/magnetization gradient is sufficiently large to cause a \mathbf{B}_{SO} that can result in a measurable torque. Here, at least an order-of-magnitude estimate of \mathbf{B}_{SO} is required.

Re: We sincerely appreciate the reviewer for the valuable suggestion. According to Zhang's theoretical work, a new torque (also called Zhang-Li torque) is induced by the nonuniform distribution of magnetization [Phys. Rev. Lett. 93, 127204 (2004)], which is also verified by experiments [Nat. Mater. 16, 712 (2017); Phys. Rev. B 102, 214408 (2020); Sci. Rep. 9, 9592 (2019)]. In our case, the SAW attenuation in the z -direction induces a Zhang-Li torque $\boldsymbol{\tau} = -\frac{c}{M_s} \mathbf{M} \times \frac{\partial \mathbf{M}}{\partial z}$ and the spin-orbit field $\mathbf{B}_{SO} = -\frac{c}{M_s} \frac{\partial \mathbf{M}}{\partial z}$, where the coefficient c depends on effective interfacial spin Hall angle θ_{SH}^{eff} . As reviewer's comment, the parameter c makes it difficult to estimate \mathbf{B}_{SO} . In previous work that we discussed in Q1, we find that they estimate the value of c [Nat. Mater. 16, 712 (2017)]. In their work, the x -direction effective magnetic field \mathbf{B}_{SO} arising from the Zhang-Li torque can realize the field-free switching of spin-orbit torque (SOT) device with perpendicular anisotropy. Because the critical switching current density depends on the x -direction magnetic field, they can smartly estimate the value of the \mathbf{B}_{SO} . In their work, they also define $\mathbf{B}_{SO} = -\frac{c}{M_s} \frac{\partial M_x}{\partial x}$. To match their experimental data with the macro-spin simulation, they obtain the value $c = 4.8 \times 10^{-5} \text{ Tm}$, where $B_{SO} = -0.26 \text{ mT}$, $\frac{\partial M_x}{\partial x} = 4.3 \times 10^6 \text{ A/m}^2$ and $M_s = 8.0 \times 10^5 \text{ A/m}$ for Co/Ni/Co. Because Pt/Ni system is similar with Pt/Co/Ni/Co/Pt system, we will use their c value to estimate the order-of-magnitude of \mathbf{B}_{SO} . In our case, the magnetization gradient $\frac{\partial M}{\partial z} = 2.9 \times 10^{10}$

A/m^2 and $M_s = 4.85 \times 10^5 \text{ A/m}$ for Ni. Therefore, the spin-orbit field $B_{\text{SO}} = -\frac{c}{M_s} \frac{\partial M}{\partial z} = -2.9 \text{ T}$. The order-of-magnitude of \mathbf{B}_{SO} should be Tesla, which is large enough to induce the acoustic spin rotation. We are sorry that we do not provide a perfect way to directly measure the c value of Zhang-Li torque here, which is a remaining weak point of the manuscript.

Please kindly check the quantitative estimation of \mathbf{B}_{SO} in the revised manuscript (lines 230-234, on page 12).

Questions of the Third Reviewer

Q1. I think the authors have addressed the questions raised by the referees satisfactorily. Experimental details and the origin of the observed phenomena are well explained. I recommend the publication of the manuscript.

Re: We sincerely appreciate the reviewer for the recommendation and previous constructive suggestions that greatly enhance the quality of our manuscript.

Reviewers' Comments:

Reviewer #2:

Remarks to the Author:

The authors have fully resolved my remaining questions and I recommend the manuscript for publication.